**Diversity and distribution of Nitrogen Fixation Genes in the Oxygen Minimum Zones of the**
**World Oceans**
[1]**Amal Jayakumar and** [1]**Bess B. Ward**
[1]Department of Geosciences
Princeton University
Princeton, NJ 08544
**Abstract**
Diversity and community composition of nitrogen fixing microbes in the three main oxygen
minimum zones (OMZs) of the world ocean were investigated using operational taxonomic unit
(OTU) analysis of *nifH* clone libraries.  Representatives of the all four main clusters of *nifH* genes
were detected.  Cluster I sequences were most diverse in the surface waters and the most abundant
OTUs were affiliated with Alpha- and Gammaproteobacteria.  Cluster II, III, IV assemblages were
most diverse at oxygen depleted depths and none of the sequences were closely related to sequences
from cultivated organisms.  The OTUs were biogeographically distinct for the most part – there was
little overlap among regions, between depths or between cDNA and DNA.  Only a few
cyanobacterial sequences were detected.  The prevalence and diversity of microbes that harbour *nifH*
genes in the OMZ regions, where low rates of N fixation are reported, remains an enigma.
**Introduction**

22         Nitrogen fixation is the biological process that introduces new biologically available

nitrogen into the ocean, and thus constrains the overall productivity of large regions of the ocean
where N is limiting to primary production.  The most abundant and most important diazotrophs
in the ocean are cyanobacteria, members of the filamentous genus *Trichodesmium* and several
unicellular genera, including *Chrocosphaera sp*. and the symbiotic genus *Candidatus*
Atelocyanobacterium thalassa (UCYN-A).  Although these cyanobacterial species are wide
spread and have different biogeographical distributions (Moisander et al. 2010), they are
restricted to surface waters, mainly in tropical or subtropical regions.

Because diazotrophs have an ecological advantage in N depleted waters, and because those

conditions occur in the vicinity of oxygen minimum zones, due to the loss of fixed N by
denitrification, it has been proposed that N fixation should be favoured in regions of the ocean
influenced by OMZs (Deutsch et al. 2007). It has also been suggested that the energetic constraints
on N fixation might be partially alleviated under reducing, i.e., anoxic, conditions (Großkopf and
LaRoche 2012).  In response to these ideas, the search for organisms with the capacity to fix
nitrogen has been focused recently in regions of the ocean that contain OMZs.  That search usually
takes the form of characterizing and quantifying one of the genes involved in the fixation reaction,
*nifH*, which encodes the dinitrogenase reductase enzyme.  Diverse *nifH* assemblages have been
reported from the oxygen minimum zone of the Eastern Tropical South Pacific (Turk-Kubo et al.
2014, Loescher et al. 2016, Fernandez et al. 2011) and the Costa Rica Dome, at the edge of the OMZ
in the Eastern Tropical North Pacific (Cheung et al 2016).   The search for non cyanobacterial
diazotrophs has resulted in discovery of diverse *nifH* genes, but they have not been associated with
significant rates of N fixation (Moisander et al. 2017).  Here we report on the distribution and
diversity of *nifH* genes in all three of the world ocean's major OMZs, including samples from both
surface and anoxic depths, and both DNA and cDNA (i.e., both presence and expression of the *nifH*
genes).

**Materials and Methods:**
Samples analysed for this study were collected from the three major OMZ regions of the
world oceans (32 total samples, Table1 and 2) from surface, oxycline and oxygen depleted zone
(ODZ) depths. Particulate material from water samples (5 – 10 L), collected using Niskin samplers,
mounted on a CTD (Conductivity-Temperature-Depth) rosette system (Sea-Bird Electronics), was
filtered onto Sterivex capsules (0.2 µm filter, Millipore, Inc., Bedford, MA) immediately after
collection using peristaltic pumps. The filters were flash frozen in liquid nitrogen and stored at -
80°C until DNA and RNA could be extracted. For samples from the Arabian Sea, DNA extraction
was carried out using the PUREGENE$^{TM}$ Genomic DNA Isolation Kit (Qiagen, Germantown, MD)
and the RNA was extracted using the ALLPrep DNA/RNA Mini Kit (Qiagen, Germantown, MD).
For samples collected from ETNP and ETSP DNA and RNA were simultaneously extracted using
the ALLPrep DNA/RNA Mini Kit (Qiagen, Germantown, MD). SuperScript III First Strand
Synthesis System (Invitrogen, Carlsbad, CA, USA) was used to synthesise cDNA immediately after
extraction following purification of RNA using the procedure described by the manufacturer,
including RT controls. DNA was quantified using PicoGreen fluorescence (Molecular Probes,
Eugene, OR) calibrated with several dilutions of phage lambda standards.
PCR amplification of *nifH* genes from environmental sample DNA and cDNA was done on
an MJ100 Thermal Cycler (MJ Research) using Promega PCR kit following the nested reaction
(Zehr et al. 1998), with slight modification as in Jayakumar et al. (2017). Briefly,  25µl PCR
reactions containing 50 pmoles each of outer primer and 20-25ng of template DNA, were amplified
for 30 cycles  (1 min at 98°C, 1 min at 57°C, 1 min at 72°C), followed by amplification with the
inner PCR primers 50 pmoles each  (Zehr and McReynolds 1989). Water for negative controls and
PCR was freshly autoclaved and UV-irradiated every day.  Negative controls were run with every
PCR experiment, to minimize the possibility of amplifying contaminants (Zehr et al. 2003). The
PCR preparation station was also UV irradiated for 1 hour before use each day and the number of
amplification cycles was limited to 30 for each reaction. Each reagent was tested separately for
amplification in negative controls. *nifH* bands were excised from PCR products after electrophoresis
on 1.2% agarose gel, and were cleaned using a QIAquick Nucleotide Removal Kit (Qiagen).  Clean
*nifH* products were inserted into a pCR®2.1-TOPO® vector using One Shot® TOP10 Chemically
Competent *E. coli*, TOPO TA Cloning® Kit (Invitrogen) according to manufacturer's specifications.
Inserted fragments were amplified with M13 Forward (-20) and M13 Reverse primers from
randomly picked clones.  PCR products were sequenced at Macrogen DNA Analysis Facility using
Big Dye^TM terminator chemistry (Applied Biosystems, Carlsbad, CA, USA). Sequences were edited
using FinchTV ver. 1.4.0 (Geospiza Inc.), and checked for identity using BLAST. Consensus *nifH*
sequences (359 bp) were translated to amino acid (aa) sequences (108 aa after trimming the primer
region) and aligned using ClustalX (Thompson et al. 1997) along with published *nifH* sequences
from the NCBI database.  Neighbor-joining trees were produced from the alignment using distance
matrix methods (PAUP 4.0, Sinauer Associates). Bootstrap analysis was used to estimate the
reliability of phylogenetic reconstruction (1000 iterations).  The *nifH* sequence from *Methanosarcina*
*lacustris* (AAL02156) was used as an outgroup. The accession numbers  from GenBank for the *nifH*
sequences in this study are Arabian Sea DNA sequences JF429940‑ JF429973 and cDNA sequences
accession numbers JQ358610‑JQ358707,  ETNP DNA sequences KY967751-KY967929 and cDNA
sequence KY967930-KY968089, and ETSP DNA sequences MK408165‑MK408307 and cDNA
sequences MK408308‑MK408422.

The *nifH* nucleotide alignment (of 787 sequences) was used to define operational

taxonomic units (OTUs) on the basis of DNA sequence identity.  Distance matrices based on this
nucleotide alignment were generated in MOTHUR (Schloss and Handlesman 2009).  The
relative *nifH* richness within each clone library was evaluated using rarefaction analysis.  OTUs
were defined as sequences which differed by ≤3% using the furthest neighbor method in the
MOTHUR program (Schloss and Handlesman 2009).  The 3% OTU definition is similar to the
level at which species are conventionally defined using 16S rDNA sequences, so it may
overestimate the meaningful diversity of the functional gene.  Redundancy analysis was
performed in R using the vegan package.  Environmental variables were transformed using
decostand.

**Results and Discussion:**
DNA and cDNA sequences (787 in total) derived from the OMZ regions of the Arabian
Sea (AS), Eastern Tropical North Pacific (ETNP) and Eastern Tropical South Pacific (ETSP)
were subjected to OTU  and phylogenetic analyses to compare the diversity and community
composition, biogeography and gene expression, of *nifH* possessing microbes among the three
OMZ regions. Phylogenetic analysis of the sequences from the AS, ETNP and ETSP were
reported previously (Jayakumar et al. 2012, Jayakumar et al. 2017, Chang et al. 2019), but the
sequences have been combined for additional analyses here. We compared the threshold OTU
definitions at 3 and 10% and found that the number of OTUs decreased, as expected, as the
resolution decreased.  Even at the 3% threshold, however, OTUs tended to separate by depth and
location, indicating a functionally useful distinction at this level. Thresholds of 3 – 5% as the
OTU definition correspond to within and between species level distinctions for *nifH* (Gaby et al.
2018).  The sequences from the OMZ regions represented all four sequence clusters (I, II, III, IV)
described by Zehr et al. (1998).

**Cluster I *nifH* OTU distributions:**  Diversity analysis of the *nifH* cluster 1 sequences
for the three OMZs based on OTUs using MOTHUR identified 41 OTUs at a distance threshold
of 3% (Supplemental Table 1A and B).  The number of sequences and the number of OTUs
varied widely among depths and stations, so the results are grouped by region (AS, ETNP,
ETSP) or depth horizon (surface or OMZ, including upper oxycline depths) or cDNA vs DNA
(Table 2). The OTUs are numbered in order of decreasing abundance in the clone library, i.e.,
OTU-1 was the most common OTU.

For all regions and depths combined, the number of OTUs detected (41) was less than the

sum of OTUs detected when each region was analyzed separately (45), indicating that there was
some overlap of OTUs among regions. The overlap was not large, however. Only three of the 12
most abundant OTUs contained sequences from more than one region and none contained
sequences from all three regions (Figure 1A).  When sequences for all three regions were
combined, only four of the 12 most abundant OTUs contained sequences from both depth
horizons (Figure 1B). Most OTUs represented a single depth, and many a single sample.
Interestingly, Cheung et al. (2016) used 454-pyrosequencing to obtain a similar number of OTUs
(37 total) from the Costa Rica Dome, and all of the 15 samples investigated by Cheung et al.
(2016) were dominated (>50%) by one of five major OTUs.

The Arabian Sea was strikingly less diverse than other regions and sample subsets

(Figure 2).  For example, when all DNA and cDNA sequences for all depths are grouped
together, the Arabian Sea (OTUs = 14, Chao = 21) contains less species richness than the
combined surface samples from all three regions (OTUs = 25, Chao = 52), despite having a
similar number of total sequences (178 for the Arabian Sea, 198 for all surface samples
combined).  This lack of diversity in the AS data may be partly due to the preponderance of
cDNA sequences, which generally contained less diversity than a similar number of DNA
sequences (see below).
Although similar numbers of sequences were obtained for cDNA (255) vs DNA (257),
the OTU "density", i.e., number of OTUs per number of sequences analyzed, was higher for
DNA (0.136 for DNA, 0.094 for cDNA).  The Chao statistic verified this observation for the
combined data from each region in predicting higher total numbers of OTUs for DNA (Chao =
42) than for cDNA (Chao = 24).  This difference could indicate that some of the *nifH* genes
present were not expressed at the time of sampling, but the cDNA sequences were not simply a
subset of the DNA community. Half of the 12 most abundant OTUs contained either cDNA or
DNA (Figure 1C), meaning that some genes were never expressed and some expressed genes
could not be detected in the DNA.
For all regions combined, similar numbers of OTUs were detected in surface waters
(OTUs = 25) and in OMZ samples (OTUs = 23), although a larger number of sequences was
analyzed for the OMZ environment (198 vs. 314 sequences for surface and OMZ depths,
respectively).  It might be expected that the presence of phototrophic diazotrophs in the surface
water would lead to greater diversity there, but only one OTU representing a known
cyanobacterial phototroph (OTU-12 = *Katagmynene spiralis* or *Trichodesmium*) was identified,
so most of the additional diversity must be present in heterotrophic or unknown sequences.
Rarefaction curves (Figure 2) indicate that sampling did not approach saturation either for
region or depth.  The Chao statistic also indicated that much diversity remains to be explored,
despite the great uncertainty in these estimates.  The total number of OTUs detected, the shape of
the rarefaction curve and the diversity indicators (Figure 2, Table 2) all indicate that the greatest
*nifH* diversity occurred in surface waters, and much of that diversity was in singletons, i.e., not
represented in the 12 most abundant OTUs, which represented 441 (86 %) of the total 512 *nifH*
Cluster 1 sequences analyzed.  Most of that diversity was contained in the ETNP, not solely a
function of number of sequences analyzed (Figure 2).

**Cluster I *nifH* Phylogeny:**  Phylogenetic affiliations at both DNA and protein level are

shown for the 12 most abundant OTUs in Table 3. The most abundant OTU (129 sequences),
OTU-1, contained Gammaproteobacterial DNA and cDNA sequences from both surface and
OMZ depths of the ETNP and cDNA sequences from oxycline and OMZ depths in the Arabian
Sea (Figure 3).  Although very similar to each other, none of these sequences had higher than
91% identity at the DNA level (96% at AA level) with cultivated strains and were most closely
related to *Pseudomonas stutzeri*.  *P. stutzeri* is a commonly isolated marine denitrifier, but it is
also known to possess the capacity for N fixation (Krotzky and Werner 1987). OTU-4, OTU-6
and OTU-8 also contained Gammaproteobacterial sequences.   All had high identity with
cultivated strains at the protein level but none were >91% identical to cultivated strains at the
DNA level.

Gammaproteobacterial sequences with very close identities to *Azotobacter vinelandii* have

been reported from the Arabian Sea ODZ and also from the ETSP (Turk-Kubo et al. 2014). This
group of *nifH* sequences with close identities to *A. vinelandii* was also retrieved from the English
Channel, Himalayan soil, South Pacific gyre, Gulf of Mexico, mangrove soil and many other
environments (Figure 3). *Azotobacter*- like sequences were included in OTU-6 but were not closest
identity at the DNA level.  Although a large number of clones were analyzed here, no sequence that
was closely associated with *A. vinelandii* was retrieved from the three regions. None of the g-
244774A11 sequences, Gammaproteobacterial relatives that were abundant in the South Pacific
(Moisander et al. 2014), were detected in this study.

OTUs-2, 3, 5, 10, and 11 all represented Alphaproteobacterial sequences, with closest

identities to various *Bradyrhizobium, Sphingomonas* and *Methylosinus* species.  Thus,
Alphaproteobacterial sequences (206 sequences) were the most abundant in the clone library. OTU-2
contained almost exclusively ETSP ODZ DNA and cDNA sequences (plus one AS ODZ DNA
sequence). OTU-3 contained DNA sequences from ETNP surface waters. OTU-5 contained
exclusively Arabian Sea DNA sequences from Station 3, while OTU-10 contained only surface
samples from the ETNP.  An OTU threshold of 11% grouped all (179 sequences in five OTUs) of
these Alphaproteobacterial sequences together, but the 3% threshold is consistent with the
phylogenetic tree, which shows small scale biogeographical separation of sequence groups.

OTUs-7 and -9 were identified as Betaproteobacteria with closest identities to *Rubrivivax*

*gelatinosum* and *Burkholderia*, 91 and 90% respectively at the DNA level. However, at the AA
level, these sequences were 99 and 100% identical to *Novosphingobium malaysiense* and *S.*
*azotifigens,* both Alphaproteobacteria, and again were biogeographically distinct.  OTU-7 contained
25 DNA sequences from the ODZ depths in the Arabian Sea, and OTU-9 contained 17
*Burkholderia*-like sequences from the oxycline at Station 1 in the Arabian Sea.  No
Betaproteobacterial *nifH* sequences were detected in the ETNP or ETSP, but sequences similar to
*Burkholderia phymatum*, *Cupriavidus sp.* and *Sinorhizobium meliloti* were reported from ETSP
previously (Fernandez et al. 2015).  Consistent with our previous report, however, there is no clear
separation between the alfa and the beta groups in *nifH* phylogeny (Jayakumar et al 2017).

Most of the Cluster I ETSP sequences from this study were contained in two OTUs (2 and 4).

OTU-2 contained 89 Alphaproteobacterial sequences with >98% identity to *nifH* sequences from
*Bradyrhizobium* sp. Uncultured bacterial sequences retrieved from the South China Sea, English
Channel, mangrove sediment, wastewater treatment and grassland soil were related to these ETSP
sequences.  OTU-4 contained 29 Gammaproteobacterial sequences retrieved from both surface and
ODZ depths.   Four of the remaining ETSP Cluster I sequences were grouped together as OTU-17
(Alphaproteobacteria, 89 and 96% identities with *Methyloceanibacter sp.* and *Bradyrhizobium sp.* at
the DNA and AA level respectively), three were in OTU-23 (*Bradyrhizobium* 100% identity) and
two were singletons.  One of the singletons was most closely related to uncultured soil and sediment
sequences and to *Azorhizobium sp*. (86%) and one had 97% identity with *Bradyrhizobium*
*denitrificans* and many sequences from marine sediments.
OTU-22 represents the Deltaproteobacterial group.  This novel group was reported
previously from the ETNP (Jayakumar et al. 2017) and has three sequences from Arabian sea (OTU-
22) and two singletons from ETNP surface waters. *nifH* possessing Deltaproteobacteria have been
reported not only from all the three ODZs but also in several other marine environments including
Chesapeake Bay water column, microbial mats from intertidal sandy beach in a Dutch barrier island,
Jiaozhou Bay sediment, Rongcheng Bay sediment, Bohai Sea, Mediterranean Sea, Narragansett Bay,
and the south Pacific gyre.
Proteobacteria-like sequences are the most frequently reported *nifH* sequences from the
OMZs studied here and similar environments.  Thirty one of 37 OTUs detected by Cheung et al
(2016) in the Costa Rica Dome OMZ were Proteobacteria, the two most common OTUs being
closely related to Alphaproteobacterium *Methylocella palustris* and the Gammaproteobacterium
*Vibrio diazotrophicus*.  Loescher et al. (2014, 2016) also found *V. diazotrophicus*-like sequences, as
well as several other Gammaproteobacteria in the ETSP.  *V. diazotrophicus* was reported previously
in the Arabian Sea (Jayakumar et al. 2012) but was not prominent in the present study.
*Bradyrhizobium spp.*, one of the most common genera reported here and in surface waters of the
Arabian Sea (Bird and Wyman 2013) and by Fernandez et al. (2011) in the ETSP, were also detected
in the Costa Rica Dome OMZ and were the dominant OTU at 1000 m at one station (Cheung et al.
2016).  In addition to *Bradyrhizobium*-like and *Teredinibacter*-like *nifH* sequences, Turk-Kubo et al.
(2014) found four other abundant Gammaproteobactera-like *nifH* sequences, which were entirely
novel.  The "Gamma A", which are commonly reported non-cyanobacteria diazotroph *nifH*
sequences from non-OMZ environments (Langlois et al. 2015, Moisander et al. 2017), were
represented by a singleton from the ETNP in the present study.
*nifH* sequences related to various Alphaproteobacterial methylotrophs are commonly found
in OMZs:  *Methylosinus trichosporium*-like sequences, which are reported here in OTU-5 from the
Arabian Sea at both surface and ODZ depths, were also reported by Fernandez et al. (2011) in the
ETSP.  *Methylocella palustris*-like *nifH* genes comprised the most common OTU in the ODZ core
depths in the Costa Rica Dome (Cheung et al. 2016). *M trichosporium* and *M. palustris* represent
obligate and facultative methanotrophs, respectively, both also obligately aerobic.  Detection of *nifH*
genes closely related to those of methanotrophs does not prove that methanotrophy is present or
important in the anoxic environment of the ODZ but the consistency of this finding across sites
motivates further investigation on the potential for methane production and consumption in ODZs.
The pattern of high diversity of *nifH*-bearing mostly heterotrophic microbes, but dominance
in each sample by one or a small number of *nifH* OTUs, suggests a bloom and bust pattern of
organic matter-supported growth.  That is, we suggest that organic matter, which is supplied
episodically in the upwelling regimes, stimulates the growth of copiotrophic microbes that respond
rapidly in bloom like fashion.  This bloom scenario has been described for denitrifying bacteria
based on the OTU patterns observed in the *nirS* and *nirK* genes as a function of the stage of
denitrification in both natural assemblages and incubated samples from OMZs (Jayakumar et al.
2009). The role of *nifH* in these heterotrophic microbes is unclear, especially because rates of
nitrogen fixation in these locations in the absence of cyanobacteria is often very low (Turk-Kubo et
al. 2014, Loescher et al. 2016, Chang et al. 2019).

Although *Trichodesmium*-like clones have been retrieved from the surface waters of the

Arabian Sea and the ETNP OMZs, only ten clones (OTU-12) in the combined clone library analyzed
here were related to *Trichodesmium* (98% identity), including both cDNA and DNA from the
Arabian Sea and cDNA from the ETNP. These sequences were actually 100% identical to
*Katagnymene spiralis*, a close relative of *Trichodesmium* isolated from the South Pacific Ocean.
Turk-Kubo et al. (2014) also retrieved only a few cyanobacterial sequences from the ETSP. No other
cyanobacterial *nifH* sequences were identified.
**Clusters II, III, IV *nifH* OTU distributions:** The other three *nifH* clusters were combined
for OTU analysis due to the limited number of sequences and OTUs obtained. A total of 18 OTUs
were identified in the combined set of 275 sequences with a 3% distance threshold (Table 3). Most
of the Cluster II, III, IV sequences were from the ETNP and ETSP. As with the Cluster I sequences,
there was very little geographic and depth overlap among these OTUs (Figure 4A, 4B). Only OTU-
1 contained sequences from more than one site, the ETNP and the ETSP. OTU-2 contained only
cDNA sequences representing ODZ depths at both ETNP stations. OTU-3 contained exclusively
ETSP DNA sequences from surface and cDNA sequences from ODZ depths. Only 10 of the Cluster
II, III, IV sequences were from the Arabian Sea, and they formed three separate OTUs, a greater
"OTU density" than was present at either of the Pacific sites. As observed for Cluster I, most of the
OTUs that were detected in the DNA were not being expressed, and those that were expressed were
not detected in the DNA (Figure 4C).
Rarefaction curves (Figure 5) indicate that sampling for Cluster II, III, IV did not
approach saturation.  The Chao statistic also indicated that much diversity remains to be
explored, despite the great uncertainty in these estimates.  Unlike the Cluster I analysis, there
were relatively few singletons in the Cluster II, III, IV data and the assemblages were dominated
by a few types.
**Clusters II, III, IV *nifH* phylogeny:**  Three large OTUs (OTU-1, -4 and -6) in Clusters II,
III, IV belonged to *nifH* Cluster IV and Alphaproteobacteria/Spirochaeta and Deltaproteobacteria
were the dominant phylogenies (Table 3, Figure 6).  The largest OTU, OTU-1, contained 88 DNA
sequences from the ETNP ODZ depths from both stations and from both depths in the ETSP. This
OTU had no similarity to any cultured microbe.  OTU-4 contained 30 sequences from the ETSP, all
cDNA from one surface station, in *nifH* Cluster IV.
OTU-2 (75 sequences) in Cluster II contained only cDNA sequences, all from ODZ
samples in the ETNP (both stations), and had no close relatives among cultivated species.  Turk-
Kubo et al. (2014) also retrieved a few clones identified as belonging to Cluster II from the
euphotic zone of the ETSP. OTU-3 contained 35 sequences in Cluster III and was dominated by
DNA sequences from surface depths of the ETSP.  OTU-5 represented Deltaproteobacteria in
*nifH* Cluster III and contained 18 identical DNA sequences from 90 m at Station BB1 in the
ETNP. Thus, of the five most common OTUs (89% of the total Cluster II, III, IV sequences
analyzed), only one could be identified to a closely related genus (i.e., OTU-4 with 90% identity
with *R. palustris*) and there was no overlap between DNA and cDNA OTUs from the same
depths.
The other 13 OTUs in the Cluster II, III, IV sequences represented either Cluster III or IV.
None of these were very closely related to any cultivated sequences. OTU-6 contained both DNA
and cDNA from the OMZ at one ETSP station.  OTU-7 contained four sequences from ETNP
surface waters with close identities with a sequence retrieved from Bohai sea.  OTU-11, had one
DNA and one cDNA sequences from the ETSP. All of the other sequences were less than 84%
identical to any sequence in the database and could only be loosely identified as Firmicutes or
Proteobacteria.

Although there were few high identities with known species, many of the Cluster II, III, IV

sequences (OTUs -2, -5, -7, -9, -10) were most closely affiliated with sulfate reducing clades at
either the DNA or protein level.  Four OTUs with highest identity to known sulfate reducers were
reported by Cheung et al. (2016) and one of them comprised nearly 40% of the sequences in one
anoxic sample.  *nifH* sequences that cluster with *Desulfovibrio spp.* are often reported from ODZ
samples (Turk-Kubo et al. 2014, Loescher et al. 2014,  Fernandez et al. 2011).  Consistent reports of
*nifH* genes associated with obligate anaerobes involved in sulfate reduction suggests a role for this
metabolism in the ODZ, again motivating further research on the significance of both sulfate
reduction and associated N fixation in ODZ waters.

**Biogeography and Environmental Correlations:**  The dominant factor determining OTU

composition and distribution is clearly biogeography (Figure 4).  That geographical factor is also
evident in the redundancy analysis (Figure 7). (Only sites that contained sequences from one of the
top OTUs are represented in the plots, so the number of site symbols is less than 32 for both plots.)
For example,  Cluster I OTU-5 containing only Arabian Sea surface sequences was positively
correlated with both T and S and all of the Arabian Sea samples clustered in the quadrant associated
with high T and S (Figure 7A).  Surface samples from the ETSP were also in that quadrant, but
surface ETNP samples were negatively correlated with S.  The surface ETNP samples correlated
with OTUs-3. -6, -10 and -11, all of which contained exclusively surface samples.  The two largest
Cluster I OTUs were associated with the  deep samples from the ETNP and ETSP and correlated
positively with nitrite concentration and negatively with oxygen – a signature of the ODZ.  Nitrate
concentration and depth did not increase the power of the analysis and were omitted from the Cluster
I RDA.  Most of the sites and five of the most common Cluster I OTUs were not well differentiated
by any of the usual environmental parameters.

The Arabian Sea contained very few sequences in Clusters II, III, IV and none of them were

in the top six OTUs, so only ETNP and ETSP samples are represented in the RDA for these clusters
(Figure 7B).  The two largest OTUs in Clusters II, III, IV  were negatively correlated with T and S
but separated along the second RDA axis, demonstrating opposite relationships with oxygen, nitrite,
and nitrate concentrations.  OTU-1 included ETSP surface sequences, as well as ODZ sequences
from both ETNP and ETSP, while OTU-2 contained only ODZ sequences but both OTUs were
phylogenetically related to anaerobic clades (Table 2).  Inclusion of all six environmental variables
was necessary to obtain maximum separation of the sites and OTUs for Clusters II, III, IV.

**Conclusions**

The OMZ regions of the world ocean contain substantial *nifH* diversity, both in surface

waters and oxygen depleted intermediate depths.  Surface waters contained greater diversity for
Cluster I, but the ODZ held the highest diversity for Clusters II, III, IV.  Cyanobacterial sequences
were rare in the combined dataset and were not detected in the ETSP.  The ETSP contained the least
diversity of Cluster I sequences, while Cluster II, III, IV were least abundant and least diverse in the
Arabian Sea.  Most of the sequences in all four Clusters of the conventional *nifH* phylogeny were not
closely related to any sequences from cultivated Bacteria or Archaea.  The most abundant OTUs in
Cluster I and in Clusters II, III, and IV could be assigned to the Alphaproteobacteria, followed by the
Gammaproteobacteria for Cluster I and Deltaproteobacteria accounted for Clusters II, III, IV
sequences.  Most of the OTUs were not shared among regions, depths or DNA vs cDNA and
sometimes were restricted to individual samples.  Some Cluster I sequences had high identity to
known species (e.g., *Bradyrhizobium*, *Trichodesmium*) but most of the Cluster II, III, IV sequences
were only distantly related to any cultured species.

The assemblage composition of *nifH*-bearing microbes is mainly explained by region, but

OTU composition was also consistent with the influence of key environmental parameters such as
oxygen and temperature, and reflects association with the secondary nitrite maximum for deep
samples.   Most of the sites/depths, both in this study and in others from OMZ regions, are
dominated by one or a few OTUs, which suggests bloom-type dynamics within a diverse
background assemblage.  Microbes occupying very similar niches and present at low population
levels might respond differentially to episodic inputs of organic matter, resulting in spatially and
temporally varying dominance by a few clades.  Thus we find similar metabolic types represented
across all the OMZs, although the specific species and genus level affiliations differ.  The consistent
detection of *nifH* sequences related to those found in known sulfate reducers and methanotrophs
suggests the need for further investigation of these pathways in ODZs.

While measurements of $N_2$ fixation rates are not reported here, the abundance of cDNA

sequences suggests that the cells harboring these genes are active.  Low, but analytically significant,
rates have been detected in ODZ depths in the ETNP (Jayakumar et al. 2017) and ETSP (Chang et
al. 2019), which suggests that non-cyanobacterial N fixation could make a minor contribution to the
nitrogen budget of the ocean.  It is therefore important in future work to determine how the diversity
described here actually contributes to biogeochemically significant reactions and what
environmental and biotic factors might influence or control the activity of diazotrophs in the dark
ocean.


**Figure Legends**

Figure 1.  Histogram of the 12 most common OTUs from Cluster I *nifH* clone libraries from the three OMZ regions.   OTUs were considered common if the total number of sequences in an OTU was ≥2% of the total number of *nifH* clones analyzed (The common OTUs contained 441 of the 512 Cluster I sequences). OTUs were defined according to 3% nucleotide sequence difference using the furthest neighbor method.  OTU designation is from most common (OTU-1) to least.  A) OTU distribution among regions.  B) OTU distribution between OMZ (including core of the ODZ and the upper oxycline depths) and surface depths (oxygenated water).  C) OTU distribution of cDNA vs DNA clones.

Figure 2.  Rarefaction curve displaying observed OTU richness versus the number of clones sequenced for Cluster I *nifH* sequences (cDNA and DNA). OTUs were defined and designated as in Figure 1.  Chao estimators (individual symbols) are shown for each of the same subsets represented in the rarefaction curves.

Figure 3.  Phylogenetic tree of Cluster 1 based on amino acid sequences.  Positions of the OTUs are  shown relative to their nearest neighbors from the database.  Individual sequence identities comprising each OTU are listed in Supplemental Table 2.

Figure 4.  Histogram of the 6 most common OTUs from Cluster II, III, IV *nifH* clone libraries from the three OMZ regions.   OTUs were considered common if the total number of sequences in an OTU was ≥2% of the total number of *nifH* clones analyzed (the common OTUs contained

252 of the 275 Cluster II, III, IV sequences). OTUs were defined according to 3% nucleotide
sequence difference using the furthest neighbor method.  OTU designation is from most common
(OTU-1) to least.  A)  OTU distribution among regions.  B)  OTU distribution between OMZ
(including core of the ODZ and the upper oxycline depths) and surface depths (oxygenated
water).  C)  OTU distribution of cDNA vs DNA clones.

Figure 5. Rarefaction curve displaying observed OTU richness versus the number of clones
sequenced for Cluster II, III, IV *nifH* sequences (cDNA and DNA). OTUs were defined and
designated as in Figure 4.  Chao estimators (individual symbols) are shown for each of the same
subsets represented in the rarefaction curves.

Figure 6.  Phylogenetic tree of Clusters II, III, IV based on amino acid sequences.  Positions of
the OTUs are  shown relative to their nearest neighbors from the database.  Individual sequence
identities comprising each OTU are listed in Supplemental Table 2.

Figure 7.  RDA plots for (A) Cluster I and (B) Clusters II, III, IV illustrating the relationships
among OTUs (green circles) and sites.  DNA = squares; cDNA = circles.  Arabian Sea = cyan
(surface) and blue (OMZ); ETNP = pink (surface) and red (deep); ETSP = yellow (surface) and
orange (deep). (A) Twelve most abundant OTUs for Cluster I and the four most independent
environmental variables. T = temperature, S = salinity, NO2 = nitrite concentration, O2 = oxygen
concentration.  (B) Six most abundant OTUs for Clusters II, III, IV and all six environmental
variables. NO3 = nitrate concentration, Z = depth.


**Tables**

Table 1.  Sampling regions and depths and sequences derived from each depth

Table 2.  OTU summary for both clusters

Richness and diversity statistics for *nifH* clone libraries from three OMZ regions.  ACE and

Chao are non-parametric estimators that predict the total number of OTUs in the original sample.

Table 3.  OTU identities for both clusters

Cultivated species with closest nucleotide identity to the OTUs identified in the *nifH* clone

libraries from three OMZ regions.  Only the 12 most common OTUs (out of 41 total) are listed

for Cluster 1 sequences, and the six most common (out of 18 total) for the Clusters II, III, IV

libraries.

Supplemental

S Table 1A and B.  List of sequences in each OTU for both clusters

S Table 2

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

Figure.1

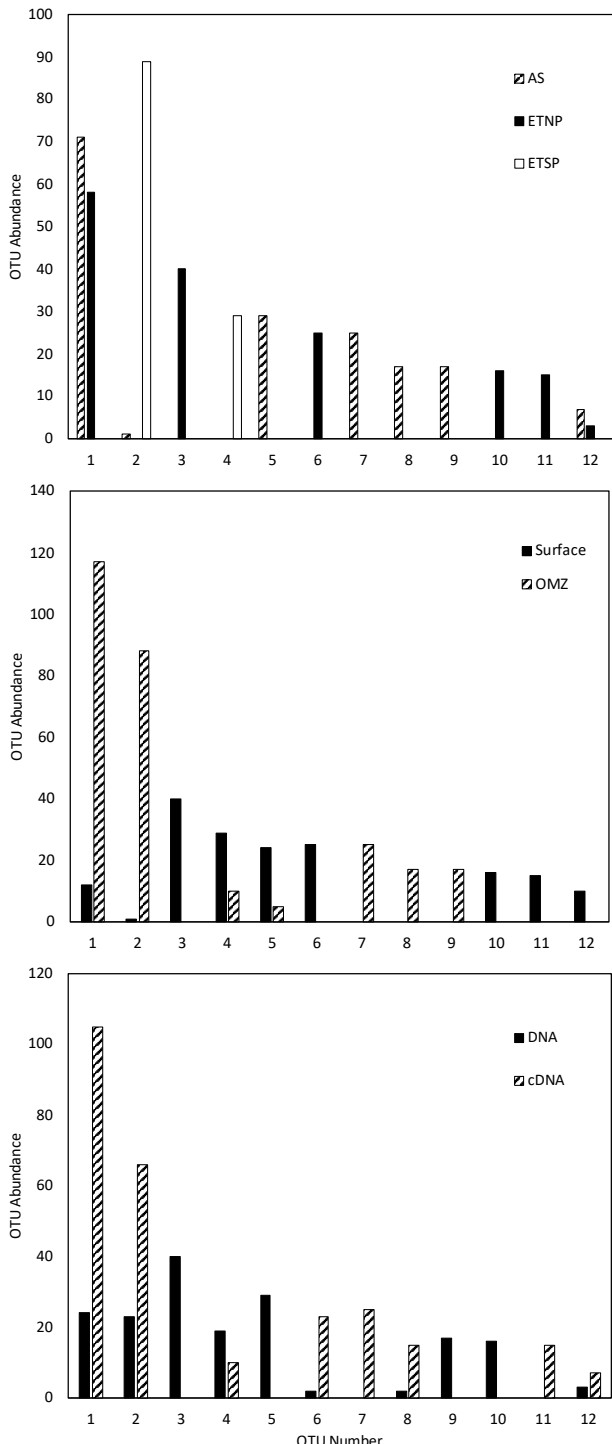


Figure. 2

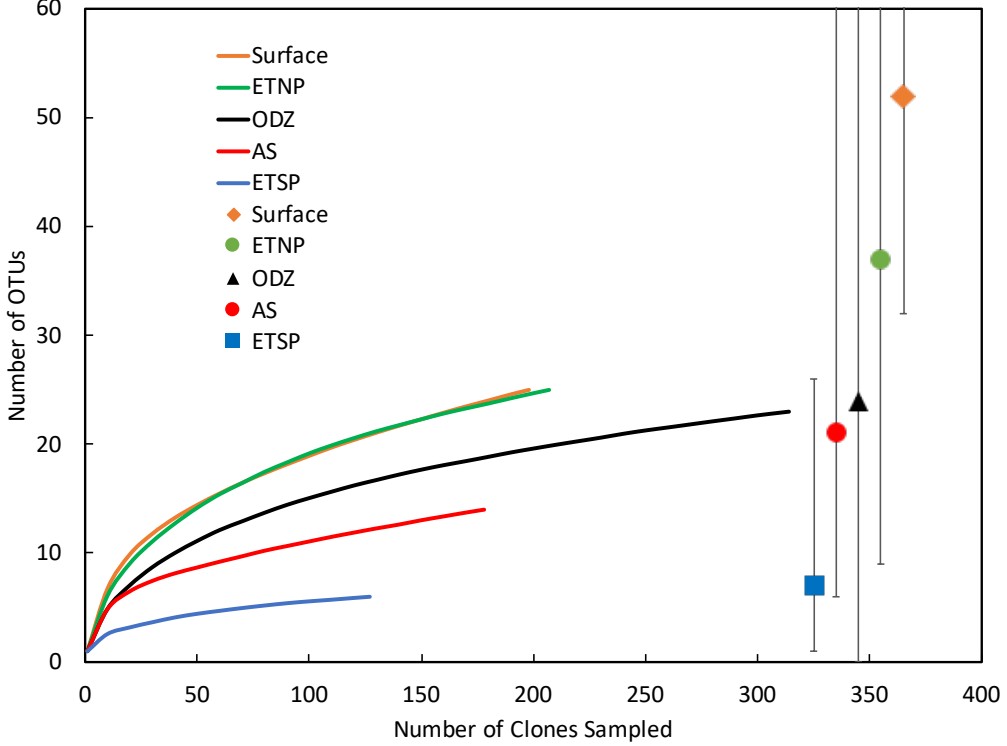

Figure. 3

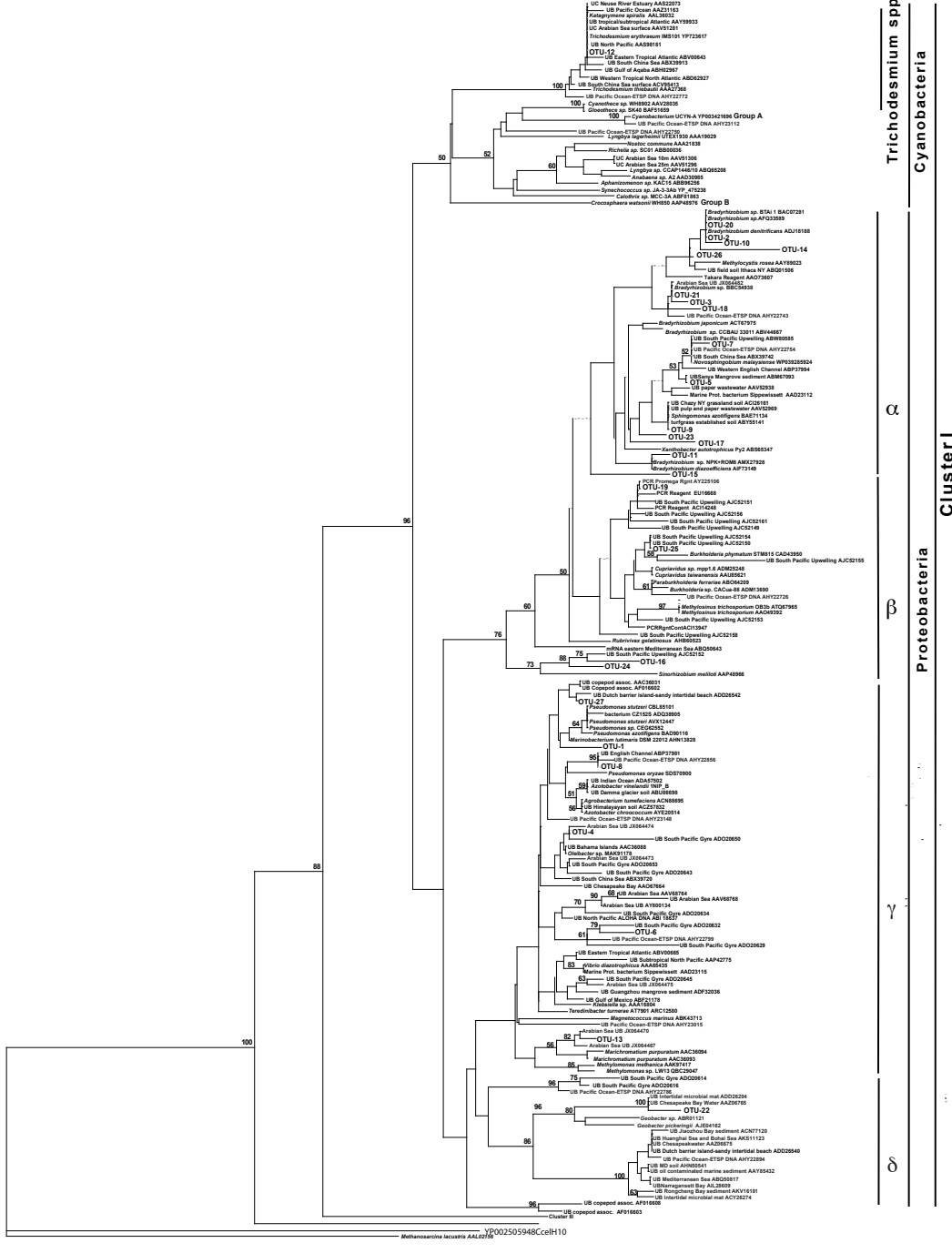

— 0.01 changes

Figure. 4

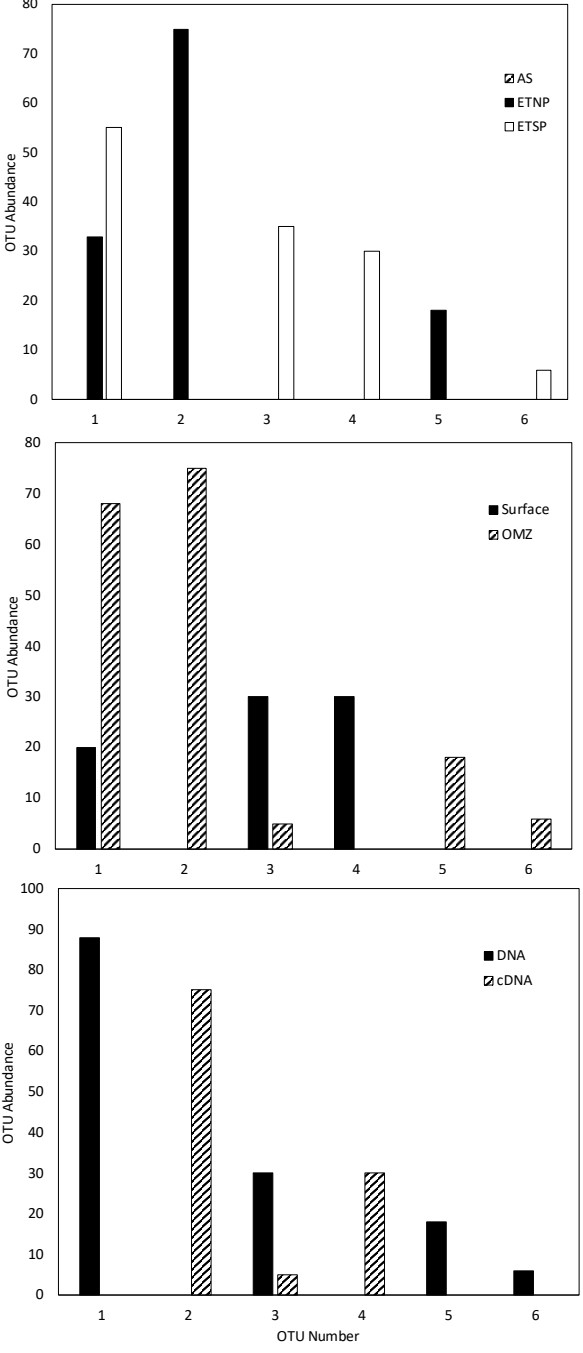


Figure. 5.

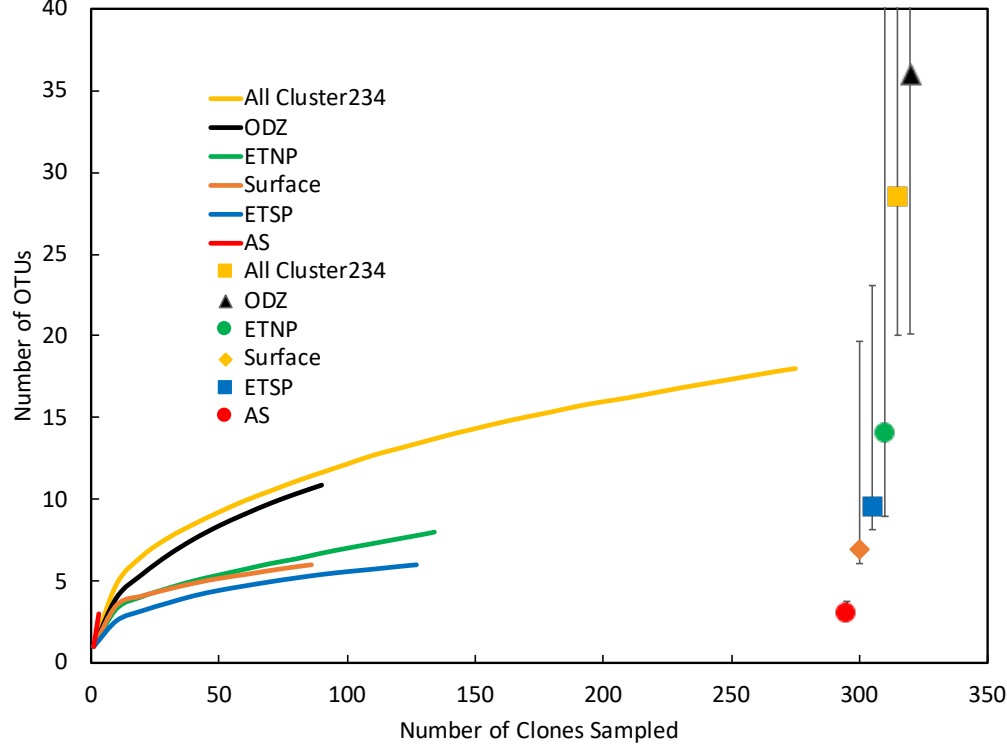


Figure 6.

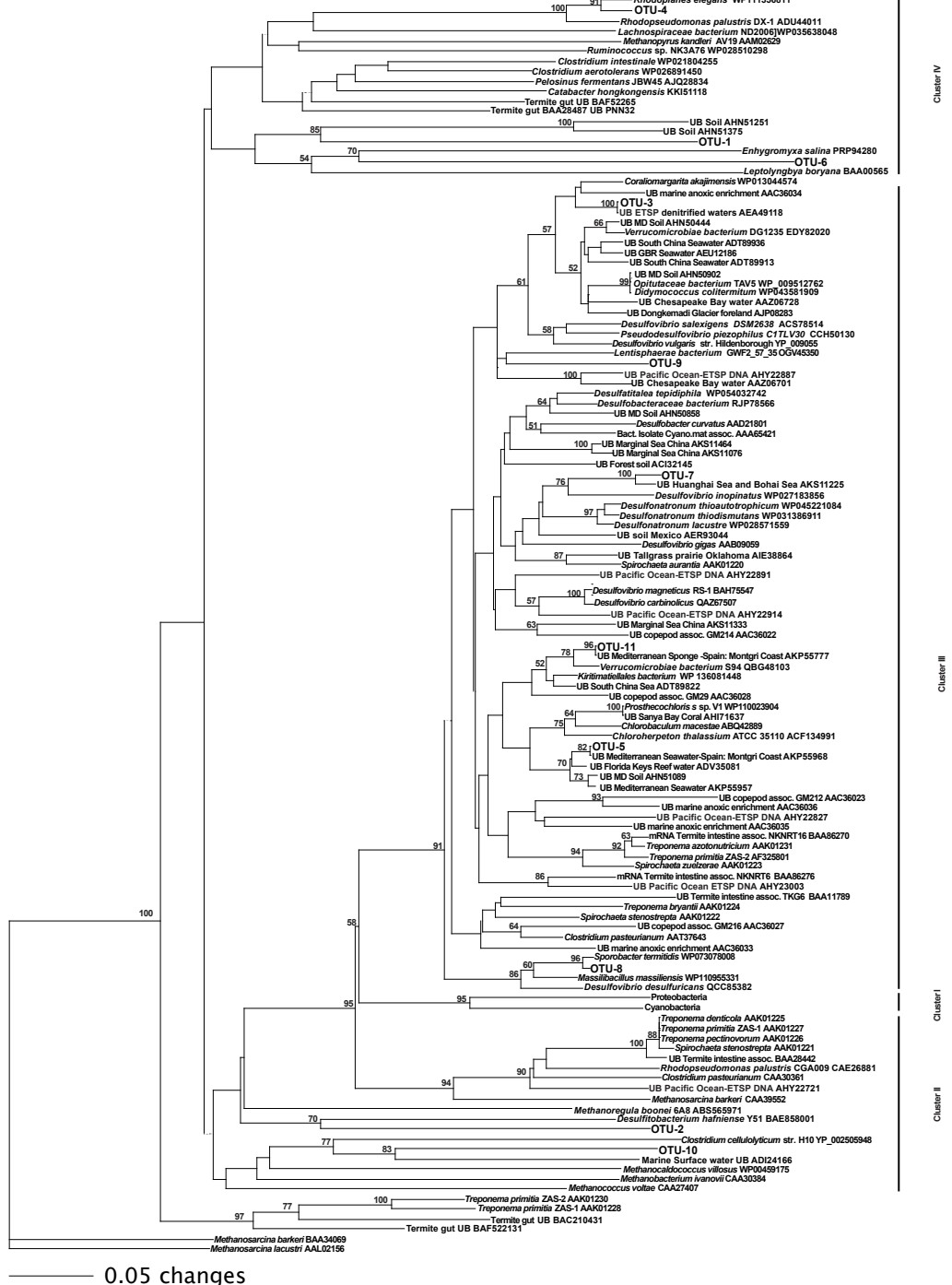

# Figure 7A

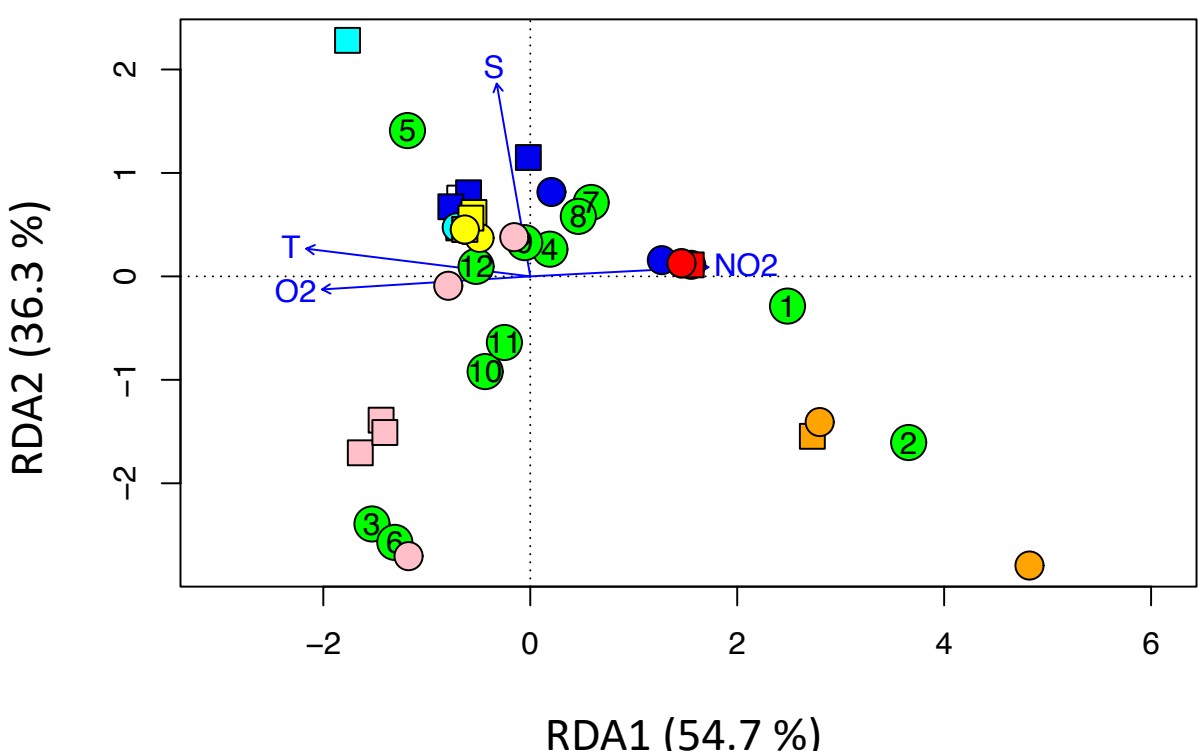

# Figure 7B

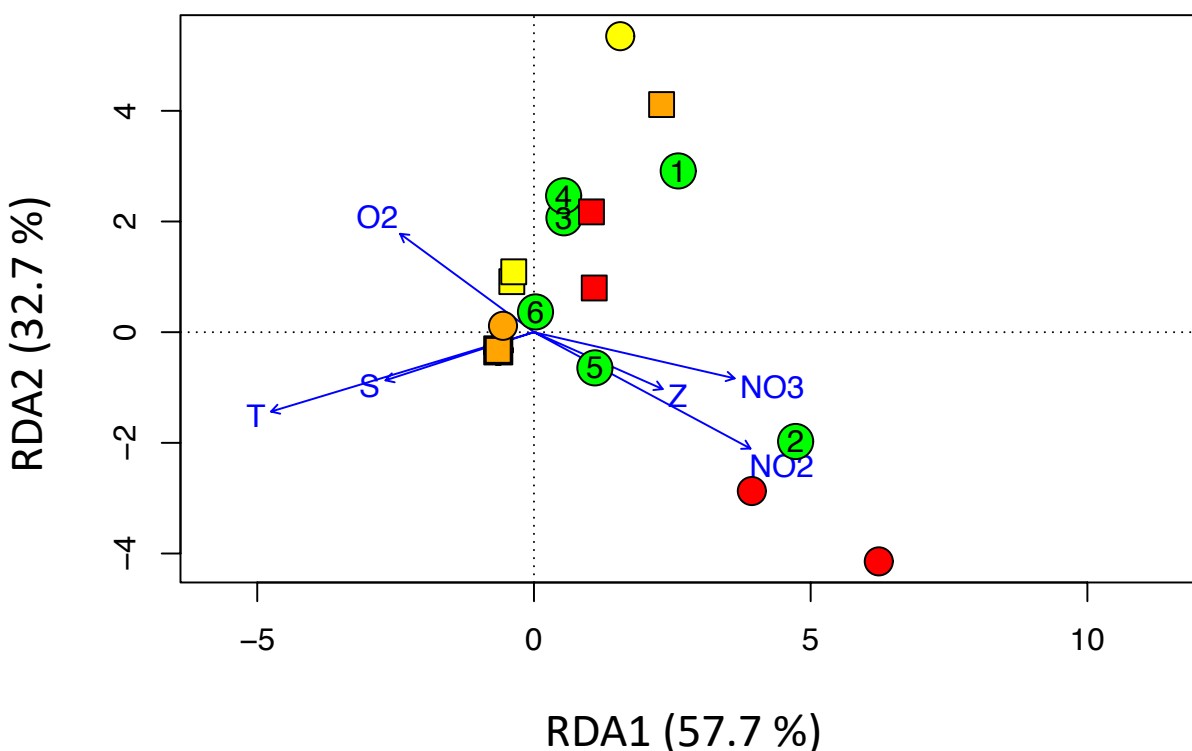

Table 1 Sampling position details

| OMZ Region | Station | Latitude | Longitude | Depth (m) | DNA Seqs | cDNA Seqs |
|---|---|---|---|---|---|---|
| Arabian Sea | S1 | 19°N | 67°E | 10 | 3 | 0 |
| Arabian Sea | S1 | 19°N | 67°E | 60 | 20 | 0 |
| Arabian Sea | S1 | 19°N | 67°E | 150 | 23 | 25 |
| Arabian Sea | S1 | 19°N | 67°E | 175 | 10 | 22 |
| Arabian Sea | S2 | 15°N | 64°E | 150 | 4 | 25 |
| Arabian Sea | S3 | 12°N | 64°E | 10 | 25 | 4 |
| Arabian Sea | S3 | 12°N | 64°E | 110 | 4 | 23 |
| ETNP | BB1 | 20 9.6°N | 106°W | 0 | 26 | 5 |
| ETNP | BB1 | 20 9.6°N | 106°W | 18 | 24 | 17 |
| ETNP | BB1 | 20 9.6°N | 106°W | 90 | 42 | 38 |
| ETNP | BB2 | 16 31°N | 107 6.8°W | 0 | 40 | 35 |
| ETNP | BB2 | 16 31°N | 107 6.8°W | 150 | 47 | 67 |
| ETSP | BB1 | 13 59.9°S | 81 12.0°W | 2 | 29 | 1 |
| ETSP | BB1 | 13 59.9°S | 81 12.0°W | 130 | 46 | 44 |
| ETSP | BB2 | 20. 46.1°S | 70 39. 5°W | 20 | 45 | 30 |
| ETSP | BB2 | 20. 46.1°S | 70 39. 5°W | 115 | 23 | 40 |


Table 2 OTU Summary

| Sample subset Cluster I | Depths, regions included | No. of Sequences | No. of Unique Sequences | No. of OTUs (cutoff ~3) | OTU /seq | Shannon | Simpson | Chao | Ace |
|---|---|---|---|---|---|---|---|---|---|
| AS | Arabian Sea, all depths | 178 | 36 | 14 | 0.079 | 1.8 | 0.22 | 21 | 45 |
| ETNP | ETNP, all depths | 207 | 80 | 25 | 0.121 | 2.37 | 0.14 | 37 | 34 |
| ETSP | ETSP, OMZ depths | 127 | 51 | 6 | 0.047 | 0.87 | 0.53 | 7 | 8 |
| All ClusterI | Three regions, all depths | 512 | 165 | 41 | 0.080 | 2.7 | 0.11 | 59 | 67 |
| All ClusterI DNA | Three regions, all depths | 257 | 97 | 35 | 0.136 | 2.8 | 0.08 | 42 | 45 |
| All ClusterI cDNA | Three regions, all depths | 255 | 75 | 24 | 0.094 | 1.7 | 0.25 | 24 | 27 |
| All ClusterI Surface | Three regions, surface depths | 198 | 73 | 25 | 0.126 | 2.5 | 0.10 | 52 | 75 |
| All ClusterI OMZ | Three regions, all depths | 314 | 98 | 23 | 0.073 | 0.9 | 0.23 | 30 | 37 |
| **Clusters II, III, IV** | | | | | | | | | |
| AS | Arabian Sea, all depths | 10 | 6 | 3 | 0.300 | 1.09 | 0.27 | 3 | 3 |
| ETNP | ETNP, all depths | 134 | 49 | 8 | 0.060 | 1.19 | 0.39 | 14 | 38 |
| ETSP | ETSP, all depths | 131 | 64 | 8 | 0.061 | 1.37 | 0.30 | 9 | 19 |
| All Clusters II,III,IV | Three regions, all depths | 275 | 117 | 18 | 0.065 | 1.88 | 0.21 | 28 | 26 |
| All Clusters II,III,IV DNA | Three regions, all depths | 155 | 65 | 12 | 0.077 | 1.20 | 0.37 | 22 | 17 |
| All Clusters II,III,IV cDNA | Three regions, all depths | 120 | 56 | 9 | 0.075 | 1.11 | 0.45 | 12 | 15 |
| All Clusters II,III,IV Surface | Three regions, surface depths | 86 | 46 | 6 | 0.070 | 1.32 | 0.29 | 7 | 13 |
| All Clusters II,III IV OMZ | Three regions, OMZ depths | 189 | 76 | 15 | 0.079 | 1.57 | 0.29 | 46 | 24 |


| | No. of Sequences | Phylogenetic Affiliation | Closest cultured relative (DNA) | Identity DNA % | Coverage % | Closest cultured relative (Protein) | Identity AA % | Coverage % |
|---|---|---|---|---|---|---|---|---|
| **Cluster I** | | | | | | | | |
| OTU-1 | 129 | Gamma | *Psuedomonas stutzeri* | 91 | 98 | *Pseudomonas stutzeri strain SGAir0442* | 95.8 | 99 |
| OTU-2 | 89 | Alpha | *Bradyrhizobium sp* | 99 | 100 | *Bradyrhizobium denitrificans strain LMG 8443* | 99 | 99 |
| OTU-3 | 40 | Alpha | *Bradyrhizobium sp. TM124* | 94 | 98 | *Bradyrhizobium sp. MAFF 210318* | 99 | 98 |
| OTU-4 | 29 | Gamma | *Marinobacterium lutimaris* | 87 | 100 | *Oleibacter sp* | 100 | 99 |
| OTU-5 | 29 | Alpha | *Methylosinus trichosporium* | 92 | 99 | *Sphingomonas azotifigens* | 99 | 100 |
| OTU-6 | 25 | Gamma | *Azotobacter chroococcum strain B3* | 81 | 99 | *Psuedomonas stutzeri* | 94 | 99 |
| OTU-7 | 25 | Beta/Alpha | *Rubrivivax gelatinosus* | 91 | 99 | *Novosphingobium malasiense* | 99 | 100 |
| OTU-8 | 17 | Gamma | *Psuedomonas stutzeri* | 91 | 98 | *Azotobacter chroococcum* strain B3 | 97 | 100 |
| OTU-9 | 17 | Beta/Alfa | *Burkholderia* | 90 | 100 | *Sphingomonas azotifigens* | 100 | 100 |
| OTU-10 | 16 | Alpha | *Bradyrhizobium* | 97 | 98 | *Bradyrhizobium sp. ORS 285* | 99 | 99 |
| OTU-11 | 15 | Alpha | *Bradyrhizobium* | 97 | 98 | *Bradyrhizobium diazoefficiens* | 98 | 99 |
| OTU-12 | 10 | Cyanobacterium | *Katagnymene spiralis* | 100 | 99 | *Trichodesmium erythraeum* | 100 | 99 |
| | | | | | | | | |
| **Clusters II, III IV** | | | | | | | | |
| OTU-1 | 88 | Alpha/Spirochaetaceae | *Rhizobium sp* | 74 | 59 | *Treponema primitia ZAS-1]* | 55 | 98 |
| OTU-2 | 75 | Delta/Firmicutes | *Geobacter* | 73 | 43 | *Desulfitobacterium hafniense* | 98 | 61 |
| OTU-3 | 35 | Verrumicrobia | *Opitutaceae bacterium* | 82 | 99 | *Coraliomargarita akajimensis* | 95 | 99 |
| OTU-4 | 30 | Alpha | *Rhodopseudomonas palustris* | 90 | 98 | *Rhodoplanes elegans* | 96 | 99 |
| OTU-5 | 18 | Delta/Chlorobi | *Desulfovibrio piezophilus* | 79 | 99 | *Prosthecochloris sp. V1, Chloroherpeton thalassium, Chloroherpeton thalassium* | 92 | 99 |
| OTU-6 | 6 | Beta/Delta | *Azoarcus communis* | 70 | 88 | *Enhygromyxa salina* | 70 | 74 |
| OTU-7 | 4 | Delta | *Desulfovibrio carbinolicus strain DSM 3852* | 81 | 99 | *Desulfovibrio inopinatus* | 90 | 99 |
| OTU-8 | 4 | Delta/Firmicutes | *Desulfovibrio desulfuricans strain IC1* | 77 | 100 | *Sporobacter termitidis* | 99 | 99 |

| OTU-9 | 3 | Delta/Lentisphae rae | *Desulfovibrio magneticus RS-1 DNA* | 84 | 100 | *Lentisphaerae bacterium GWF2_57_35, Desulfatitalea tepidiphila, Desulfobacteraceae bacterium* | 84 | 100 |
| OTU-10 | 3 | Delta/Methanoc occi | *Desulfovibrio desulfuricans strain IC1* | 77 | 100 | *Methanocaldococcus villosus* | 65 | 99 |
| OTU-11 | 2 | Verrucomicrobi a | Verrucomicrobia bacterium S94 | 87 | 100 | Verrucomicrobia bacterium S94 | 97 | 99 |
