# Peer review of "Diversity and distribution of Nitrogen Fixation Genes in the Oxygen Minimum Zones of the"

_Biogeosciences, 2019_

## Referee Comment (RC1) · Anonymous Referee #1 · 27 Jan 2020

The study of Jayakumar and Ward investigates the microbial nitrogen fixing community in three major oxygen minimum zones in the worlds oceans. They took samples in the ETNP, ETSP and IO where they analysed OTUs using up-to-date DNA / RNA extraction and amplification methods. The specificity of the Q-PCR assays is described in detail and the construction of phylogenetic trees is fine. All methods are described extensively and the data set allows for the first time a solid comparison among sites. Moreover, the paper is clearly written and presents interesting conclusions such as low diversity in the Arabian Sea compared to other sites. The manuscript can be published as is.

[Figure]

I would only appreciate some details like the station number and depths sampled in the material and methods section. Otherwise it is a timely and very valuable contribution to our understanding of the diversity of nitrogen fixing microbes in the ocean.

―――――――――――――――――――――――

---

## Referee Comment (RC2) · Anonymous Referee #2 · 2 Mar 2020

In this study, the authors have reported the diversity and phylogeny of putative diazotrophs in the three major OMZs of the Ocean, including ETNP, ETSP and Arabian Sea. By analysing the clone libraries of nifH gene fragments derived from DNA and RNA samples (787 sequences in total), the authors compared the putative diazotroph communities in surface, oxycline and oxygen depleted water of the three OMZs. Basically, my major concerns are the significance of the finding and validity of the approach in this study. It should be noted that the phylogenetic diversity of putative diazotrophs in these three major OMZs have been reported detailedly in previous studies (Jayakumar et al., 2012 & 2017; Loescher et al., 2014; Cheung et al., 2016). Therefore, I afraid that the current study does not provide significant amount of new knowledge to the

field. For the approach, 787 nifH sequences are definitely insufficient to reconstruct the diazotroph communities in different layers of the three OMZs. The authors can calculate the coverage indices to evaluate whether the sequencing depths are enough. With such limited dataset, I doubt if it is meaningful and convincing to compare the diversity and community composition of putative diazotrophs in different waters. The authors stated that most of the OTUs were not shared among the regions (L 297), while it could also be the result of limited sequencing depth. Given that this study is mainly about phylogenetic diversity, the authors should consider using high-throughput next generation sequencing of nifH amplicons to provide more convincing findings.

Specific comments:

1) L30: The authors should briefly talk about the previous studies about the putative diazotrophs in OMZs, including the works done by other teams.

2) L45: Please provide detailed information about the sampling locations and depths.

3) L89-111: Details of qPCR assay were listed in the methodology, while the relevant result was not mentioned at all.

4) L123: Please specify the sequence number of each sample.

5) L137: The diverse nifH phylotypes of 4 different clusters and their affiliated strains have already been discussed in the previous studies. Is there any the new finding worth elaborating? How about the correlation between nifH phylotypes and environmental variables?

6) L281: How about the other stations? How many stations in total? It may be easier to follow if the authors show the diaoztroph community composition in each station clearly.

―――――――――――――――――

---

## Author Comment (AC1) · 8 May 2020

The study of Jayakumar and Ward investigates the microbial nitrogen fixing community in three major oxygen minimum zones in the worlds oceans. They took samples in the ETNP, ETSP and IO where they analysed OTUs using up-to-date DNA / RNA extraction and amplification methods. The specificity of the Q-PCR assays is described in detail and the construction of phylogenetic trees is fine. All methods are described extensively and the data set allows for the first time a solid comparison among sites. Moreover, the paper is clearly written and presents interesting conclusions such as low diversity in the Arabian Sea compared to other sites. The manuscript can be published

as is.

I would only appreciate some details like the station number and depths sampled in the material and methods section. Otherwise it is a timely and very valuable contribution to our understanding of the diversity of nitrogen fixing microbes in the ocean.

Response: Thank you very much for your constructive comments, we now have Inserted new table with the information
* * *
[Figure]

Table 1 Sampling position details

| OMZ Region | Station | Latitude | Longitude | Depth (m) | DNA Seqs | cDNA Seqs |
|---|---|---|---|---|---|---|
| Arabian Sea | S1 | 19°N | 67°E | 10 | 3 | 0 |
| Arabian Sea | S1 | 19°N | 67°E | 60 | 20 | 0 |
| Arabian Sea | S1 | 19°N | 67°E | 150 | 23 | 25 |
| Arabian Sea | S1 | 19°N | 67°E | 175 | 10 | 22 |
| Arabian Sea | S2 | 15°N | 64°E | 150 | 4 | 25 |
| Arabian Sea | S3 | 12°N | 64°E | 10 | 25 | 4 |
| Arabian Sea | S3 | 12°N | 64°E | 110 | 4 | 23 |
| ETNP | BB1 | 20 9.6°N | 106°W | 0 | 26 | 5 |
| ETNP | BB1 | 20 9.6°N | 106°W | 18 | 24 | 17 |
| ETNP | BB1 | 20 9.6°N | 106°W | 90 | 42 | 38 |
| ETNP | BB2 | 16 31°N | 107 6.8°W | 0 | 40 | 35 |
| ETNP | BB2 | 16 31°N | 107 6.8°W | 150 | 47 | 67 |
| ETSP | BB1 | 13 59.9°S | 81 12.0°W | 2 | 29 | 1 |
| ETSP | BB1 | 13 59.9°S | 81 12.0°W | 130 | 46 | 44 |
| ETSP | BB2 | 20. 46.1°S | 70 39. 5°W | 20 | 45 | 30 |
| ETSP | BB2 | 20. 46.1°S | 70 39. 5°W | 115 | 23 | 40 |

**Fig. 1.** Table 1. Sampling regions and depths and sequences derived from each depth

---

## Author Comment (AC2) · 8 May 2020

In this study, the authors have reported the diversity and phylogeny of putative diazotrophs in the three major OMZs of the Ocean, including ETNP, ETSP and Arabian Sea. By analysing the clone libraries of nifH gene fragments derived from DNA and RNA samples (787 sequences in total), the authors compared the putative diazotroph communities in surface, oxycline and oxygen depleted water of the three OMZs. Basically, my major concerns are the significance of the finding and validity of the approach in this study. It should be noted that the phylogenetic diversity of putative diazotrophs in these three major OMZs have been reported detailedly in previous studies (Jayakumar

et al., 2012 & 2017; Loescher et al., 2014; Cheung et al., 2016). Therefore, I afraid that the current study does not provide significant amount of new knowledge to the field. For the approach, 787 nifH sequences are definitely insufficient to reconstruct the diazotroph communities in different layers of the three OMZs. The authors can calculate the coverage indices to evaluate whether the sequencing depths are enough. With such limited dataset, I doubt if it is meaningful and convincing to compare the diversity and community composition of putative diazotrophs in different waters. The authors stated that most of the OTUs were not shared among the regions (L 297), while it could also be the result of limited sequencing depth. Given that this study is mainly about phylogenetic diversity, the authors should consider using high-throughput next generation sequencing of nifH amplicons to provide more convincing findings.

Response: Thank you very much for your detailed constructive comments. We are addressing the specific comments below. In previous studies, what has been reported, describes the distribution of phylotypes and rates of dinitrogen fixation in the ETNP and ETSP, no rates for nitrogen fixation were made for the Arabian Sea. In this study we are comparing the distribution of the phylotypes of nitrogen fixing microbes and their expression, amongst the three major OMZs and also between the oxic and anoxic depths and their biogeography. We agree that number of sequences obtained from some of the depths are low compared to what could be obtained using next generation sequencing on new samples (which are not available for the Arabian Sea). True, the rarefaction curves (Figure 2 and Figure 5) indicate that sampling did not approach saturation, and for Cluster I there were many singletons, indicating more unexplored diversity. Next generation sequencing of nifH amplicons would provide a better understanding of the distributions but that would be a future study in itself – here we are presenting the synthesis that is possible for all three locations with the currently available data.

In the revised manuscript, we have compared our results more broadly to the few other available data sets from similar locations, and that synthesis points to some useful
conclusions, which we now include in the discussion – see new text starting at L267 and L349. All the previous studies except Cheung et al. (2016) were based on clone libraries and they all, including Cheung et al (2016) find (where the data are available) 1) the same basic metabolic types and 2) dominance by a few OTUS in each sample. We interpret this pattern to reflect the presence of diverse assemblages that essentially bloom episodically in response to organic matter input. This interpretation makes sense for non-cyanobacterial clades in both clusters. The consistent detection of nifH genes from e.g., sulfate reducers and methanotrophs motivates further investigation of those metabolisms in ODZs.

Thus we think the conclusions from this study are indeed robust, and hypothesize that this new synthesis would be corroborated by more in depth sequencing.

Specific comments: 1) L30: The authors should briefly talk about the previous studies about the putative diazotrophs in OMZs, including the works done by other teams.

Response: We have introduced the other relevant studies in the introduction and incorporate comparisons with those studies in the results and discussion.

2) L45: Please provide detailed information about the sampling locations and depths.

Response: Thank you, Referee#1 also suggested this and we now have Inserted new table with the information

3) L89-111: Details of qPCR assay were listed in the methodology, while the relevant result was not mentioned at all.

Response: Removed this section from methodology

4) L123: Please specify the sequence number of each sample.

Response: The accession number ranges are now provided.

5) L137: The diverse nifH phylotypes of 4 different clusters and their affiliated strains have already been discussed in the previous studies. Is there any the new finding worth

elaborating? How about the correlation between nifH phylotypes and environmental variables?

Response: The new discussion sections starting at L L267 and L349 point out the new interpretations and ideas that are made possible by this synthesis of data from multiple sites and in comparison with previous studies. This is also highlighted in the conclusions.

6) L281: How about the other stations? How many stations in total? It may be easier to follow if the authors show the diaoztroph community composition in each station clearly.

Response: A new table 1 is now included which lists out the stations, positions and number of sequences obtained from each station and depths.

[Figure]

[Figure]

**Fig. 1.** Figure 7. RDA plots for (A) Cluster I and (B) Clusters II, III, IV illustrating the relationships among OTUs (green circles) and sites. DNA = squares; cDNA = circles. Arabian Sea = cyan (surface) a

---

## Referee Report (RR1)

**Review**

Jayakumar and Ward present an intercomparison of the diazotroph community composition of three prominent OMZs (ETNP, ETSP and Arabian Sea). The approach is used to analyse diversity is clone sequencing, which is arguably outdated nowadays and does not provide enough coverage, and the authors should acknowledge this drawback more clearly and be more cautious when interpreting their results accordingly. Nevertheless, the strength of the manuscript is the intercomparison of the three regions, which has not been addressed in previous studies. Hence, I consider that this work is of interest for the community and deserves publication, but first some minor changes are needed. The ecological role of diazotrophs in OMZs is not properly introduced, nor discussed. The results and discussion section look more like a report than a proper discussion. Finally, interpretations and conclusions need to be reformulated and acknowledge the drawbacks of the methods used. Below I provide specific comments.

**Introduction**

L26: "sp." should not be italicized.

L29: I suggest using sunlit waters or euphotic zone instead to account for the depth range where cyanobacterial diazotrophs are found in (sub)tropical waters.

L33: The hypothesis tested in Deutsch et al. 2007 has been extensively turned down by several publications in the past years (e.g. Bonnet et al., 2017; Knapp et al., 2016). Please account for this in your text. As it reads now, the reader perceives that this is a confirmed hypothesis still.

L43: But check (Bentzon-Tilia et al., 2015).

**Materials and Methods**

The methods are described in great detail. However, can the authors confirm that a DNase treatment was used and no-RT controls run?

L49: Please indicate which are the three major OMZ regions you include in your analysis. Perhaps it would fit better in the introduction, a brief description of the three regions, what they have in common, what they differ on…

L51: For no expert readers, an introduction to what ODZ means (which oxygen thresholds are considered), would be helpful.

L93-94: This is unclear. Each sequence that was 3% different from each other was considered an OTU? Can you provide a reference?

**Results and Discussion**

L109-111: The authors should make it clear that this is a compilation of previously published data.

L174: Groups that are both denitrifiers and $N_2$ fixers are rather common in low oxygen environments, including coastal sediments. A discussion on the double ecological role of these groups in OMZs would be appreciated.

L205-206: Not sure what the authors mean here, why the separation between alfa and beta was unlear?

L224: Please add references.

L251-253: This is a nice hypothesis but it is not sufficiently explained or referenced. How could this be tested? (idem for L355-357). What references do the authors have of non-cyanobacterial diazotrophs blooming upon important inputs of organic matter?

L357-359: Please add a reference at the end of this sentence.

L366: Replace N for $N_2$.

L367: Or rather for the nitrogen budget of OMZ zones…

L370: This is the first time that OMZ is included as part of the "dark ocean". No explanation or comparison with non-cyanobacterial diazotrophy in the dark ocean realm are provided in the manuscript.

**Tables**

Table 1: A reference of the base of the photic zone at each of these stations would be useful.

**References**

Bentzon-Tilia, M., Traving, S. J., Mantikci, M., Knudsen-Leerbeck, H., Hansen, J. L. S., Markager, S., & Riemann, L. (2015). Significant N2 fixation by heterotrophs, photoheterotrophs and heterocystous cyanobacteria in two temperate estuaries. *ISME Journal*, *9*(2), 273–285. https://doi.org/10.1038/ismej.2014.119

Bonnet, S., Caffin, M., Berthelot, H., & Moutin, T. (2017). Hot spot of N2 fixation in the western tropical South Pacific pleads for a spatial decoupling between N2 fixation and denitrification. *Proceedings of the National Academy of Sciences*. https://doi.org/10.1073/pnas.1619514114

Knapp, A. N., Casciotti, K. L., Berelson, W. M., Prokopenko, M. G., & Capone, D. G. (2016). Low rates of nitrogen fixation in eastern tropical South Pacific surface waters. *Proceedings of the National Academy of Sciences*, *113*(16), 4398–4403. https://doi.org/10.1073/pnas.1515641113

---

## Author Response (AR2)

Report #1
Submitted on 22 Jun 2020
Anonymous Referee #2

*In this version, the authors have compared their results with those in the previous studies, which resulted in a more thorough discussion about the nifH phylotypes in OMZs of the world oceans. The manuscript has been improved significantly with these revisions.*

*Nevertheless, I still have a major concern about the insufficient dataset in this study. According to the information in Table 1, there were few sequences per sample, some samples even have only 3-4 clones being sequenced. This problem is exactly what I was worried about and that's why I asked the authors to provide more details (Table 1) in the previous review. With such limited dataset, it is hard to say anything about the relative abundance of the OTUs or compare the nifH communities at different depth and OMZs. The authors need to either increase the sequencing depth or convince the audiences that their results can really represent "nifH phylotypes/community in the world OMZs". In the current version, it is obvious that the authors could not draw any convincing conclusions. Besides, I have some specific comments as follows.*

**Response**:
The reviewer is correct that clone libraries cannot provide the depth of sequencing we now expect, and *IF* we were performing the sequencing experiments today, we would use NGS, not clone libraries.  The fact is that the samples were collected many years ago and it is not possible to analyze them now using NGS.

We disagree, however, with the reviewer's conclusion that the clone libraries are not sufficient to yield interesting and convincing conclusions.  As far as we are aware, there is only one NGS study of nifH in an OMZ region (Cheung et al. 2016, and that one sampled a few depths in the Costa Rica Dome, not one of the major OMZs).  Thus our study is unique in being able to compare nifH community composition from all three major oceanic OMZs. Most of the other studies, in whose context we frame our results, were clone library studies, with the same limitations as our own, except that they all considered only one of the OMZs.  So they are all suboptimal, but useful and interesting in aggregate.

While the individual sequences are interesting in terms of their phylogenetic affiliation, all analysis of patterns and distributions was done on the basis of OTUs, which were determined from the aggregated sequences.  Aggregation was done in order to minimize the kind of bias that derives from small samples size, which is the concern of the reviewer.

Given the reviewer's preference for NGS data, the comparison of the present study with Cheung et al. 2016 is particularly interesting.  We analyzed 16 samples for DNA and RNA, and found a total of 59 OTUs (41 for Cluster I, 18 for Clusters II, III, IV). Cheung et al. (2016) analyzed 15 samples for DNA only and found 37 OTUs at the 95% level.  All of their samples All of their samples showed strong dominance in the community composition.  It's really striking that even with many more sequences, that study found only five abundant, i.e., dominant, types and one of them was also in the top four in our study.  The 37 OTUs from Cheung et al. do imply that deeper sequencing in our samples would result in higher numbers of OTUs. But it does not imply that we would see a different community composition in terms of structure/ dominance.

This reviewer's critique has helped us to improve and expand the comparative analysis in our manuscript. We hope the conclusions are clearer now, as well as the basis upon which they were made (see below).

*L16-17: "The OTUs were biogeographically distinct for the most part – there was little overlap among regions," Would it be a result of limited sequencing depth in your study? Based on Table 1, only the most dominant phylotypes could be detected with 3-50 sequences per sample.*

**Response:**
True, only the most dominant phylotypes would be detected at low sequencing depth – which actually adds ecological significance to the statement that there was little overlap among regions – dominance was observed in all three regions (i.e., the assemblage might be more diverse than we can evaluate but the most abundant types differed between locations). That is probably a robust finding, and we have made that explicit in the revised manuscript. That observation is supported by the only relevant NGS study (Cheung et al. 2016 as described in the preceding paragraph) as well in the other limited clone library studies which are available for comparison.

*L54: I did not see 32 samples in Table 1.*

**Response:**
We considered the analysis to include 32 samples because 16 samples were analyzed separately for RNA and DNA. We've revised it to make clear that there were 16 water samples but 30 clone libraries.

*L88: You should construct maximum-likelihood tree and choose the best-fit model using model test.*

**Response:**
We have taken that suggestion and redone the phylogenetic tree using Maximum Likelihood methods. We used the Poisson model as the one used by previous authors working with nifH (including Cheung et al. 2016). All of the major branch points are identical to the previously presented Neighbor Joining tree for Cluster I, but there were several changes for Clusters II, III, IV, which make the positions more consistent with previous work.

*L149: How can you compare community composition and biogeography with 3-50 sequences per sample?*

**Response:**
These comparisons were all made by grouping depths and stations within regions. The OTUs were defined from groups of sequences (Table 2). All the groups contained 86 – 275

sequences, with one exception: 10 sequences for Clusters II, III, IV for the Arabian Sea. None of the comparisons were made on the basis of 3- 50 sequences.

*L405-408: It seems over-speculation to me. The "similar metabolic types" was not supported by the result. You only sequenced some clones of nifH gene fragments. Also, without the actual abundance (i.e. gene copy number), you cannot make any statement about "bloom". "Most of the sites/depths, both in this study and in others from OMZ regions, are dominated by one or a few OTUs" could be due to limited sequencing depth or low diversity of nifH phylotypes in the samples.*

**Response:**
As described above, the conclusion that there are probably only a few important OTUs when the sequences are grouped by depth or region is a robust finding. Deeper sequencing would not discover new dominants, it would only make the tail of the distribution longer. Good point about the metabolic types. We have changed the wording to make it clear that the metabolic type is an assumption based on the phylogenetic affiliation of the nifH sequence (Line 410) by inserting the phrase "nifH genes associated with" similar metabolic types...

Report #2
Submitted on 30 Jun 2020
Anonymous Referee #3

Review
*Jayakumar and Ward present an intercomparison of the diazotroph community composition of three prominent OMZs (ETNP, ETSP and Arabian Sea). The approach is used to analyse diversity is clone sequencing, which is arguably outdated nowadays and does not provide enough coverage, and the authors should acknowledge this drawback more clearly and be more cautious when interpreting their results accordingly. Nevertheless, the strength of the manuscript is the intercomparison of the three regions, which has not been addressed in previous studies. Hence, I consider that this work is of interest for the community and deserves publication, but first some minor changes are needed. The ecological role of diazotrophs in OMZs is not properly introduced, nor discussed. The results and discussion section look more like a report than a proper discussion. Finally, interpretations and conclusions need to be reformulated and acknowledge the drawbacks of the methods used. Below I provide specific comments.*

**Response:**
We recognize the limitations of the clone library approach (see response to Referee #2) and have taken the advice of both referees to acknowledge that in the text and to point out where different conclusions might derive from deeper sequencing. We discuss our results in the context of all the nifH/OMZ publications we could find and have been careful not to overstate the power of the analysis.

It's not clear what more we can say about the "ecological role of diazotrophs in the OMZs". That question is one of the motivations for the work, but the analysis so far does not point to an obvious answer to why nifH genes are so prevalent or what role N fixation might play in the OMZ.

*Introduction*
*L26: "sp." should not be italicized.*

**Response:**
Done.

*L29: I suggest using sunlit waters or euphotic zone instead to account for the depth range where cyanobacterial diazotrophs are found in (sub)tropical waters.*

**Response:**
We added "sunlit" to the description but kept the word "surface" as well, because we use "surface" as category in the later analysis (L35).

*L33: The hypothesis tested in Deutsch et al. 2007 has been extensively turned down by several publications in the past years (e.g. Bonnet et al., 2017; Knapp et al., 2016). Please account for this in your text. As it reads now, the reader perceives that this is a confirmed hypothesis still.*

**Response:**
We have always been skeptical of this hypothesis. However, it was a motivating factor in the research. The sentence simply states that the Deutsch et al. proposed the idea, not that it is true.

*L43: But check (Bentzon-Tilia et al., 2015).*

**Response:**
Bentzon-Tilia et al 2015 detected N2 fixation in samples in which heterotrophic diazotrophs were abundant and were able to estimate indirectly that the heterotrophs might have contributed 20 – 50 % to one of the measured rates. That study was in two very shallow estuaries, so it's hard to know how to relate them to this work or the N fixation phenomenon in OMZs. Interestingly, however, some of the non-cyanobacterial nifH OTUs were phylogenetically very similar to the heterotrophic diazotrophs reported here and elsewhere. We can cite the Bentzon-Tilia et al. paper and replace "not" with "rarely".

*Materials and Methods*
*The methods are described in great detail. However, can the authors confirm that a DNase treatment was used and no-RT controls run?*

**Response:**
RT controls were mentioned in the text (implied no-RT) but we have made both of these steps explicit in the revised text (L 82).

*L49: Please indicate which are the three major OMZ regions you include in your analysis. Perhaps it would fit better in the introduction, a brief description of the three regions, what they have in common, what they differ on…*

**Response:**
We have added very brief descriptions of the three OMZ regions in the introduction (L55).

*L51: For no expert readers, an introduction to what ODZ means (which oxygen thresholds are considered), would be helpful.*

**Response:**
We removed this unnecessary acronym.

*L93-94: This is unclear. Each sequence that was 3% different from each other was considered an OTU? Can you provide a reference?*

**Response:**
An OTU is simply defined by some threshold cutoff in identity. There is no standard cutoff, so we looked at several cutoffs (between 3% and 10%) and settled on 3% as the one that makes phylogenetic and biogeographically meaningful distinctions. So yes, sequences that are 3% different would be in different OTUs. Useful OTU cutoffs differ among different genes (e.g., ribosomal genes are less variable than functional genes as a rule) and some functional genes are more highly conserved than others (e.g., RuBisCO is much more conserved that nitrate reductase in diatoms). We cited Schloss and Handlesman (2009) for the OTU threshold method and Gaby et al (2018) for the threshold that is meaningful for nifH.

*Results and Discussion*
*L109-111: The authors should make it clear that this is a compilation of previously published data.*

**Response:**
The papers in which they were previously published are explicitly cited in L135, previously L110.

*L174: Groups that are both denitrifiers and N$_2$ fixers are rather common in low oxygen environments, including coastal sediments. A discussion on the double ecological role of these groups in OMZs would be appreciated.*

**Response:**
True, the cooccurrence of N2 fixers and denitrifiers, even of both capabilities within the same microbe, is often observed. To comment on the ecological role or the biogeochemical significance of that finding would be pure speculation here, as we have no information to go on other than the presence of the sequences.

*L205-206: Not sure what the authors mean here, why the separation between alfa and beta was unlear?*

**Response:**
Alpha and Beta -proteobacterial are distinguished on the basis of their 16S rRNA genes, but their nifH sequences don't always follow the same distinctions. So you can't always tell from the nifH alone whether it comes from an Alpha or a Beta -proteobacterium. That must mean a combination of lateral gene transfer and evolution has occurred somewhat independently for the functional gene, which is subject to different selection than is the rRNA gene. In the phylogenetic tree, it is obvious that the nifH sequences do not fall out exactly along the branch pattern predicted from the 16S rRNA tree for the Betaproteobacteria.

*L224: Please add references.*

**Response:**
The references can be found by following the accession numbers, which are clearly presented in the tree itself. It is not usually necessary to list the citations for every sequence, when that information is associated with the unique accession numbers. Each of the sequences listed in this paragraph is identified exactly in the tree and the reader can follow the references from the accession numbers. This is the usual convention for citation because not every sequence is associated with a literature publication and not every sequence included in the tree is mentioned in the text.

*L251-253: This is a nice hypothesis but it is not sufficiently explained or referenced. How could this be tested? (idem for L355-357). What references do the authors have of noncyanobacterial diazotrophs blooming upon important inputs of organic matter?*

**Response:**
We mention it as a hypothesis, but testing it is far beyond the scope of this work. The analogy of denitrifiers was provided, and we have now added a sentence about a published study in which N2 fixation was stimulated by organic matter addition (L301).

*L357-359: Please add a reference at the end of this sentence.*

**Response:**
That sentence is just another statement of the hypothesis mentioned above. It is a suggestion derived from our own work, not a citation.

*L366: Replace N for N$_2$.*

**Response:**
Done.

*L367: Or rather for the nitrogen budget of OMZ zones…*

**Response:**
If it is a minor contribution to the N budget of the OMZ, it is certainly a minor contribution to the N budget of the ocean. It is the latter that people have been trying to address by looking for N fixation in OMZs, i.e., the ocean is the box of interest here, not the OMZ as a separate box in the oceanic inventory.

*L370: This is the first time that OMZ is included as part of the "dark ocean". No explanation or comparison with non-cyanobacterial diazotrophy in the dark ocean realm are provided in the manuscript.*

**Response:**
We removed reference to the dark ocean.

*Tables*
*Table 1: A reference of the base of the photic zone at each of these stations would be useful.*

**Response:**

Thanks for suggesting that clarification. We have added a column to Table 1 to indicate which samples were considered surface (within the euphotic zone) and which were OMZ (all of which are below the euphotic zone).

Princeton, NJ 08544

**Abstract**

Diversity and community composition of nitrogen fixing microbes in the three main oxygen minimum zones (OMZs) of the world ocean were investigated using operational taxonomic unit (OTU) analysis of *nifH* clone libraries.  Representatives of the all four main clusters of *nifH* genes were detected.  Cluster I sequences were most diverse in the surface waters and the most abundant

OTUs were affiliated with Alpha- and Gammaproteobacteria.  Cluster II, III, IV assemblages were most diverse at oxygen depleted depths and none of the sequences were closely related to sequences from cultivated organisms. The OTUs were biogeographically distinct for the most part – there was little overlap among regions, between depths or between cDNA and DNA.  In this study of all three

OMZ regions, and from the few other published reports from individual OMZ sites, dominance by a few OTUs was commonly observed.  This pattern suggests dynamic response of the components of the overall diverse assemblage to variable environmental conditions.  Community composition in most samples was not clearly explained by environmental factors, but the most abundant OTUs were differentially correlated with the obvious variables, temperature, salinity, oxygen and nitrite concentrations.   Only a few cyanobacterial sequences were detected.  The prevalence and diversity of microbes that harbour *nifH* genes in the OMZ regions, where low rates of N fixation are reported, remains an enigma.

**Introduction**

Nitrogen fixation is the biological process that introduces new biologically available nitrogen into the ocean, and thus constrains the overall productivity of large regions of the ocean where N is limiting to primary production. The most abundant and most important diazotrophs in the ocean are cyanobacteria, members of the filamentous genus *Trichodesmium* and several unicellular genera, including *Chrocosphaera* sp. and the symbiotic genus *Candidatus* Atelocyanobacterium thalassa (UCYN-A). Although these cyanobacterial species are wide spread and have different biogeographical distributions (Moisander et al. 2010), they are restricted to sunlit surface waters, mainly in tropical or subtropical regions.

Because diazotrophs have an ecological advantage in N depleted waters, and because those conditions occur in the vicinity of oxygen minimum zones, due to the loss of fixed N by denitrification, it has been proposed that N fixation should be favoured in regions of the ocean influenced by OMZs (Deutsch et al. 2007). It has also been suggested that the energetic constraints on N fixation might be partially alleviated under reducing, i.e., anoxic, conditions (Großkopf and LaRoche 2012). In response to these ideas, the search for organisms with the capacity to fix nitrogen has been focused recently in regions of the ocean that contain OMZs. That search usually takes the form of characterizing and quantifying one of the genes involved in the fixation reaction, *nifH*, which encodes the dinitrogenase reductase enzyme. Diverse *nifH* assemblages have been reported from the oxygen minimum zone of the Eastern Tropical South Pacific (Turk-Kubo et al. 2014, Loescher et al. 2016, Fernandez et al. 2011) and the Costa Rica Dome, at the edge of the OMZ in the Eastern Tropical North Pacific (Cheung et al 2016). The search for non cyanobacterial diazotrophs has resulted in discovery of diverse *nifH* genes, but they have rarely been associated with significant rates of N fixation (Moisander et al. 2017, Bentzon-Tilia et al. 2015). Thus the occurrence and diversity of putative diazotrophs in nitrogen rich aphotic waters remains unexplained.

Here we report on the distribution and diversity of *nifH* genes in all three of the world ocean's major OMZs. The two Pacific OMZs, the Eastern Tropical North (ETNP) and South (ETSP)

Pacific, are both highly productive eastern boundary regions. The ETSP is the one of the most productive regions in the world ocean and has an oxygen depleted layer of about 400 m at its greatest depth. The ETNP is less well ventilated and less productive, with an anoxic layer of more than 700 m. The third major OMZ is the Arabian Sea, which is geographically constrained to the northern Indian Ocean. It experiences an annual monsoon cycle but is permanently and stably stratified with a maximum anoxic layer of about 800 m. Both surface and anoxic depths, and both

DNA and cDNA (i.e., both presence and expression of the *nifH* genes) were investigated. The approach used here to investigate diazotroph assemblages is based on clone library analysis of *nifH*

sequences. Next generation amplicon sequencing would yield greater numbers of sequences, although it might not overcome the primer bias associated with PCR and cloning. The strength of the current study is the inclusion of similar data from all three OMZs. By comparing these results to previous studies using the same and other methods, we find robust biogeographical patterns and community structure among the non-cyanobacterial diazotroph assemblages.

**Materials and Methods:**

Samples analysed for this study were collected from the three major OMZ regions of the world oceans (16 total samples, Table1) from surface and oxygen minimum zone (OMZ, including oxycline and anoxic) depths. Particulate material from water samples (5 – 10 L), collected using

[revised manuscript text omitted]

OMZ regions. Phylogenetic analysis of the sequences from the AS, ETNP and ETSP were reported separately in previous publications (Jayakumar et al. 2012, Jayakumar et al. 2017,

Chang et al. 2019), but the sequences have been combined for additional global analyses here.

We compared the threshold OTU definitions at 3 and 10% and found that the number of OTUs decreased, as expected, as the resolution decreased.  Even at the 3% threshold, however, OTUs tended to separate by depth and location, indicating a functionally useful distinction at this level.

Thresholds of 3 – 5% as the OTU definition correspond to within and between species level distinctions for *nifH* (Gaby et al. 2018).  The sequences from the OMZ regions represented three of the four sequence clusters (I, II, III, IV) described by Zehr et al. (1998).

**Cluster I *nifH* OTU distributions:**  Diversity analysis of the *nifH* cluster 1 sequences for the three OMZs based on OTUs using MOTHUR identified 41 OTUs at a distance threshold of 3% (Table 2).  The number of sequences and the number of OTUs varied widely among depths and stations, so the results are grouped by region (AS, ETNP, ETSP) or depth horizon (surface or OMZ, including upper oxycline depths) or cDNA vs DNA (Table 2).

Grouping the sequences by depth horizon (surface or OMZ), region (AS, ETSP, ETNP) or

DNA/RNA, allows the detection of patterns that are not driven by the relatively low number of sequences obtained from some of the individual clone libraries. The OTUs are numbered in order of decreasing abundance in the clone library, i.e., OTU-1 was the most common OTU.

For all regions and depths combined, the number of OTUs detected (41) was less than the sum of OTUs detected when each region was analyzed separately (45), indicating that there was some overlap of OTUs among regions. The overlap was not large, however. Only three of the 12

most abundant OTUs contained sequences from more than one region and none contained sequences from all three regions (Figure 1A).  When sequences for all three regions were combined, only four of the 12 most abundant OTUs contained sequences from both depth horizons (Figure 1B). Most OTUs represented a single depth, and many a single sample.  This suggests a pattern of dominance, rather than evenness, in the *nifH* assemblage.  Deeper sequencing is therefore expected to discover a larger number of rare OTUs, but might not change the picture that emerges here of a small number of abundant clades.  Interestingly, Cheung et al.

(2016) reported a similar pattern of dominance based on a larger DNA sequence dataset from only one location. Using 454-pyrosequencing to obtain a similar number of OTUs (37 total)

from the Costa Rica Dome, all of the 15 samples investigated by Cheung et al. (2016) were dominated (>50%) by one of five major OTUs.

The Arabian Sea was strikingly less diverse than other regions and sample subsets (Figure 2). For example, when all DNA and cDNA sequences for all depths are grouped together, the Arabian Sea (OTUs = 14, Chao = 21) contains less species richness than the combined surface samples from all three regions (OTUs = 25, Chao = 52), despite having a similar number of total sequences (178 for the Arabian Sea, 198 for all surface samples combined). This lack of diversity in the AS data may be partly due to the preponderance of cDNA sequences, which generally contained less diversity than a similar number of DNA

sequences (see below).

Although similar numbers of sequences were obtained for cDNA (255) vs DNA (257), the OTU "density", i.e., number of OTUs per number of sequences analyzed, was higher for

DNA (0.136 for DNA, 0.094 for cDNA). The Chao statistic verified this observation for the combined data from each region in predicting higher total numbers of OTUs for DNA (Chao =

42) than for cDNA (Chao = 24). This difference could indicate that some of the *nifH* genes present were not expressed at the time of sampling, but the cDNA sequences were not simply a subset of the DNA community. Half of the 12 most abundant OTUs contained either cDNA or

DNA (Figure 1C), meaning that some genes were never expressed and some expressed genes could not be detected in the DNA. Based on a similar number of sequences from each sample (1

– 52 per sample) from the ETSP, Turk-Kubo et al. (2014) also found that DNA and cDNA

clones were differently distributed among stations; one phylotype was recovered exclusively from cDNA and only one phylotype occurred in both DNA and cDNA. The relatively low

 sequencing depth associated with clone library studies limits the sensitivity of this comparison, but it clearly shows that dominant components of the DNA and cDNA libraries frequently represent different subsets of the total assemblage.

[revised manuscript text omitted]

Gammaproteobacteria in the ETSP.  *V. diazotrophicus* was reported previously in the Arabian Sea (Jayakumar et al. 2012) but was not prominent in the present study. Sequences most similar to various *V. diazotrophicus*, other *Vibrio* species, and other Gammaproteobateria, including *P.*

*stutzeri*, were the most common non-cyanobacterial Cluster I sequences reported for the low oxygen waters of the Southern California Bight (Hamersley et al. 2011). *Bradyrhizobium spp.*, one of the most common genera reported here and in surface waters of the Arabian Sea (Bird and Wyman

2013) and by Fernandez et al. (2011) in the ETSP, were also detected in the Costa Rica Dome OMZ

and were the dominant OTU at 1000 m at one station (Cheung et al. 2016). *Bradyrhizobium*-like sequences were the most abundant among those amplified from ODZ  incubations in which the $N_2$

fixation rate was enhanced by the addition of glucose (Bonnet et al 2013).  In addition to

*Bradyrhizobium*-like and *Teredinibacter*-like *nifH* sequences, Turk-Kubo et al. (2014) found four other abundant Gammaproteobactera-like *nifH* sequences, which were entirely novel.  The "Gamma

A", which are commonly reported non-cyanobacteria diazotroph *nifH* sequences from non-OMZ

environments (Langlois et al. 2015, Moisander et al. 2017), were represented by a singleton from the

ETNP in the present study.

*nifH* sequences related to various Alphaproteobacterial methylotrophs are commonly found in OMZs:  *Methylosinus trichosporium*-like sequences, which are reported here in OTU-5 from the

Arabian Sea at both surface and ODZ depths, were also reported by Fernandez et al. (2011) in the

ETSP.  *Methylocella palustris*-like *nifH* genes comprised the most common OTU in the ODZ core depths in the Costa Rica Dome (Cheung et al. 2016). *M trichosporium* and *M. palustris* represent obligate and facultative methanotrophs, respectively, both also obligately aerobic.  Detection of *nifH*

genes closely related to those of methanotrophs does not prove that methanotrophy is present or important in the anoxic environment of the ODZ but the consistency of this finding across sites motivates further investigation of the potential for methane production and consumption in ODZs.

The pattern of high diversity of *nifH*-bearing mostly heterotrophic microbes, but dominance in each sample by one or a small number of *nifH* OTUs, suggests a bloom and bust pattern of organic matter-supported growth.  That is, we suggest that organic matter, which is supplied episodically in the upwelling regimes, stimulates the growth of copiotrophic microbes that respond rapidly in bloom like fashion.  This bloom scenario has been described for denitrifying bacteria based on the OTU patterns observed in the *nirS* and *nirK* genes as a function of the stage of denitrification in both natural assemblages and incubated samples from OMZs (Jayakumar et al.

2009).  Amino acids and glucose both stimulated $N_2$ fixation in OMZ samples from the ETSP, and

*nifH* sequences associated with Alpha- and Gammaproteobacteria, as well as Cluster III phylotypes, were found in a glucose enrichment experiment (Bonnet 
[revised manuscript text omitted]

2012, Arabian Sea;  Fernandez et al. 2011, Loescher et al. 2014, Turk-Kubo et al. 2014, all from the

ETSP) or similar environments (Cheung et al. 2016, Costa Rica Dome and Hamersley et al. 2011, hypoxic basins in the Southern California Bight). Combining those reports from individual regions, plus the new sequences from the ETNP reported here, shows that most of the sites/depths, both in this study and in others from OMZ regions, are dominated by one or a few OTUs, which suggests bloom-type dynamics within a diverse background assemblage.  Microbes occupying very similar niches and present at low population levels might respond differentially to episodic inputs of organic matter, resulting in spatially and temporally varying dominance by a few clades.  Thus we find *nifH*

sequences associated with  similar metabolic types represented across all the OMZs, although the specific species and genus level affiliations differ.  The consistent detection of *nifH* sequences related to those found in known sulfate reducers and methanotrophs suggests the need for further investigation of these pathways in ODZs.

While measurements of $N_2$ fixation rates are not reported here, the abundance of cDNA

sequences suggests that the cells harboring these genes are active.  Low, but analytically significant, rates have been detected in ODZ depths in the ETNP (Jayakumar et al. 2017) and ETSP (Chang et al. 2019), which suggests that non-cyanobacterial $N_2$ fixation could make a minor contribution to the nitrogen budget of the ocean.  It is therefore important in future work to determine how the diversity described here actually contributes to biogeochemically significant reactions and what environmental and biotic factors might influence or control the activity of diazotrophs in the OMZ.

**Figure Legends**

Figure 1. Histogram of the 12 most common OTUs from Cluster I *nifH* clone libraries from the three OMZ regions.   OTUs were considered common if the total number of sequences in an

OTU was ≥2% of the total number of *nifH* clones analyzed (The common OTUs contained 441

of the 512 Cluster I sequences). OTUs were defined according to 3% nucleotide sequence difference using the furthest neighbor method.  OTU designation is from most common (OTU-1)

to least.  A) OTU distribution among regions.  B) OTU distribution between OMZ (including core of the ODZ and the upper oxycline depths) and surface depths (oxygenated water).  C)

OTU distribution of cDNA vs DNA clones.

Figure 2.  Rarefaction curve displaying observed OTU richness versus the number of clones sequenced for Cluster I *nifH* sequences (cDNA and DNA). OTUs were defined and designated as in Figure 1.  Chao estimators (individual symbols) are shown for each of the same subsets represented in the rarefaction curves.

Figure 3.  Maximum likelihood (ML) phylogenetic tree, based on Poisson model of Cluster I, partial *nifH*  translated amino acid sequences from DNA and cDNA. Bootstrap values >50% of

1,000 replications are labeled with black circles on the branches.  Accession number of reference sequences from NCBI are provided at the end of each reference names. Positions of the OTUs are  shown relative to their nearest neighbors from the database.  Individual sequence identities comprising each OTU are listed in Table 3.

Figure 4.  Histogram of the 6 most common OTUs from Cluster II, III, IV *nifH* clone libraries from the three OMZ regions.   OTUs were considered common if the total number of sequences in an OTU was ≥2% of the total number of *nifH* clones analyzed (the common OTUs contained

252 of the 275 Cluster II, III, IV sequences). OTUs were defined according to 3% nucleotide sequence difference using the furthest neighbor method.  OTU designation is from most common (OTU-1) to least.  A)  OTU distribution among regions.  B)  OTU distribution between OMZ

(including core of the ODZ and the upper oxycline depths) and surface depths (oxygenated water).  C)  OTU distribution of cDNA vs DNA clones.

Figure 5. Rarefaction curve displaying observed OTU richness versus the number of clones sequenced for Cluster II, III, IV *nifH* sequences (cDNA and DNA). OTUs were defined and designated as in Figure 4.  Chao estimators (individual symbols) are shown for each of the same subsets represented in the rarefaction curves.

Figure 6.  Maximum likelihood (ML) phylogenetic tree, based on Poisson model, of Cluster II,

III, IV partial *nifH*  translated amino acid sequences from DNA and cDNA. Bootstrap values

>50% of 1,000 replications are labeled with black circles on the branches. Accession number of reference sequences from NCBI are provided at the end of each reference names. Positions of the

OTUs are  shown relative to their nearest neighbors from the database.  Individual sequence identities comprising each OTU are listed in Table 3.

Figure 7.  RDA plots for (A) Cluster I and (B) Clusters II, III, IV illustrating the relationships among OTUs (green circles containing the OTU number) and sites.  DNA = squares; cDNA =

[revised manuscript text omitted]